# Single cell derived mRNA signals across human kidney tumors

Matthew D. Young [1,25,26✉], Thomas J. Mitchell [1,2,3,25], Lars Custers [4,5,25], Thanasis Margaritis[4], Francisco Morales-Rodriguez[4,5], Kwasi Kwakwa[1], Eleonora Khabirova [1], Gerda Kildisiute [1], Thomas R. W. Oliver [1,2], Ronald R. de Krijger[4,6], Marry M. van den Heuvel-Eibrink [4], Federico Comitani [7], Alice Piapi [8], Eva Bugallo-Blanco[8], Christine Thevanesan[8], Christina Burke [8], Elena Prigmore[1], Kirsty Ambridge[1], Kenny Roberts [1], Felipe A. Vieira Braga[9], Tim H. H. Coorens [1], Ignacio Del Valle[8], Anna Wilbrey-Clark[1], Lira Mamanova [1], Grant D. Stewart [2,3], Vincent J. Gnanapragasam[2,3,10], Dyanne Rampling[11], Neil Sebire[12], Nicholas Coleman[2,13], Liz Hook[2,13], Anne Warren [2], Muzlifah Haniffa [1,14,15], Marcel Kool[4,16,17], Stefan M. Pfister[16,17,18], John C. Achermann [8], Xiaoling He[19,20], Roger A. Barker[19,20], Adam Shlien[7,21,22], Omer A. Bayraktar [1], Sarah A. Teichmann [1,23], Frank C. Holstege [4], Kerstin B. Meyer [1], Jarno Drost [4,5,26✉], Karin Straathof [8,11,26✉] & Sam Behjati [1,2,24,26✉]

Tumor cells may share some patterns of gene expression with their cell of origin, providing clues into the differentiation state and origin of cancer. Here, we study the differentiation state and cellular origin of 1300 childhood and adult kidney tumors. Using single cell mRNA reference maps of normal tissues, we quantify reference "cellular signals" in each tumor. Quantifying global differentiation, we find that childhood tumors exhibit fetal cellular signals, replacing the presumption of "fetalness" with a quantitative measure of immaturity. By contrast, in adult cancers our assessment refutes the suggestion of dedifferentiation towards a fetal state in most cases. We find an intimate connection between developmental mesenchymal populations and childhood renal tumors. We demonstrate the diagnostic potential of our approach with a case study of a cryptic renal tumor. Our findings provide a cellular definition of human renal tumors through an approach that is broadly applicable to human cancer.

A full list of author affiliations appears at the end of the paper.

As cancer cells evolve from normal cells, they may retain patterns of messenger RNA (mRNA) characteristic of the cell of origin. In such cases, the cancer cell transcriptome may contain information that can identify the cancer cell of origin, its differentiation state, or trajectory towards a cancer cell. It is therefore conceivable that tumor transcriptomes can be used to identify the cells from which tumors arise and test fundamental hypotheses regarding tumor's differentiation states, such as the "fetalness" of childhood tumors or the dedifferentiation of adult tumors towards a fetal state.

Single cell transcriptomics allows for a direct quantitative comparison to be made between single tumor and relevant normal cell transcriptomes. For example, single cell transcriptomes identified that a specific subtype of proximal tubular cells are the normal cell correlate of clear cell renal cell carcinoma (ccRCC) cells[1]. Such experiments can also reveal more precise information about normal cells within the tumor microenvironment. However, the high resource requirements of single cell transcriptomics preclude investigations of large patient cohorts, which are required to study rare subtypes, test the generalizability of such signals and determine associations with clinical parameters. An alternative approach is to identify the presence of single cell derived mRNA signals in bulk tumor transcriptomes, utilizing normal single cell transcriptomes as a reference. Smaller numbers of single cancer cell experiments can then be used to validate cellular signals identified.

Tumor bulk transcriptomes for most types of human cancer have been generated in the context of cancer genomics efforts of recent years, such as those conducted by the *International Cancer Genome Consortium* (ICGC) and *The Cancer Genome Atlas* (TCGA)[2,3]. Single cell reference data, generated by efforts collectively known as the *Human Cell Atlas*[4,5], have begun to provide quantitative transcriptional definitions of the normal cells that constitute the developing and mature human kidneys[1,6–10]. By combining these bulk tumor transcriptome databases with single cell reference data, we may therefore be able to identify single cell signals in bulk transcriptomes across large cohorts of kidney tumors.

Here, we study normal single cell mRNA signals in bulk kidney tumor transcriptomes ($n = 1258$; Fig. 1A, Supplementary Data 1) and validate our findings using targeted single cell experiments ($n = 10$, Fig. 1A, Supplementary Table 1). There are three central aims of our analyses. Firstly, we test the fundamental presumption that childhood renal tumors exhibit fetal cell signals whilst adult tumors dedifferentiate towards a fetal state. Next, we define for each tumor type its normal cell correlate which may represent its cell of origin and provide diagnostic cues. Finally, we explore the tumor micro-environment across different tumor types.

## Results

### An integrated single cell reference map of the kidney.
The nephron is the functional unit of the kidney and together with its associated vasculature and support cells make up the majority of kidney cells. The nephron is derived from the mesoderm and forms from a combination of mesenchymal cell populations that mature into the epithelial cells of the nephron via mesenchymal to epithelial transition (MET)[11]. To precisely define these mesenchymal populations and the populations they mature into, we created a refined fetal kidney reference map combining previously generated[1,8] and newly generated human fetal kidney single cell data (Fig. 1B, S1).

This reference revealed 4 key mesenchymal populations: mesenchymal progenitor cells (MPCs), cap mesenchyme (CM), and two populations of specialized interstitial cells: smooth muscle-like cells (ICa), and cortical stromal cells (ICb) (Fig. 1B,

S1A–C)[8,11]. The cap mesenchyme condenses on the ureteric bud and forms the tubular structures of the nephron via mesenchymal to epithelial transition. The mesenchymal cells which do not form cap mesenchyme and remain in the interstitial space form interstitial support cells for the nephron, such as mesangial cells. The final mesenchymal population, which we termed mesenchymal progenitor cells, was not present in sufficient numbers to be reported in earlier single cell transcriptomic studies of the developing kidney. These MPCs are enriched for early time points (Fig. 1C), strongly resemble mesenchymal cells in the fetal adrenal (Supplementary Fig. 1D)[12], and both populations resemble primitive mesodermal populations in the post gastrulation mouse embryo (Supplementary Fig. 1D)[13]. Developmentally, both the adrenal cortex and the kidney are derived from the same mesodermal lineage.

We combined this refined map of the developing kidney with previously generated maps of the mature kidney[1], the developing adrenal gland[12], and the post-gastrulation mouse[13] (Fig. 1A). Together these provide a complete single cell reference map of the kidney across developmental time.

### Quantification of reference cellular mRNA signals in bulk transcriptomes.
Our single cell reference map of the kidney provides a cellular mRNA signal for each population of cells. To measure the abundance of these reference cellular signals in bulk tumor transcriptomes, we devised a method that fits raw bulk mRNA counts for the entire transcriptome—not just marker genes—to a weighted linear combination of transcriptomic signals derived from reference single cell data.

A number of bulk deconvolution tools exist that aim to identify the cellular composition of bulk tissues using a single cell reference[14–16]. However, the aim of our analysis was not to identify and quantify the number of cells present in the microenvironment, but to identify the major cellular signals (or transcriptional programs) used by tumor cells. As such, we do not expect any of our single cell reference populations to exactly match the tumor cells' transcriptome. We therefore designed our method to identify the major transcriptional signals (defined using single cell data) present in bulk transcriptomic data, with the expectation that the provided reference signals would not perfectly match the transcriptomes of the cells in the tumor. We term this approach "cellular signal analysis" to differentiate it from "deconvolution analysis", the inference of the cellular composition of bulk transcriptomes.

To validate our approach we applied cellular signal analysis and published deconvolution methods, MuSiC[15] and BSeq-SC[14] to human bulk transcriptomes of known origin: purified normal leucocytes, pre-B cell leukemia, and peripheral blood mononuclear cells[17,18]. For this comparison we used a reference that combined single cell transcriptomes from peripheral blood cells with a negative control population; proximal tubular kidney cells. As proximal tubular cells are completely absent from the source material for these bulk transcriptomes, an ideal method would not assign any contribution from this population.

We first considered those bulk transcriptomes which we expected to be well described by the provided reference (e.g., bulk B cell transcriptomes should be well modeled by a B cell signal). All three methods identified the correct cellular signal in most cases (Fig. 1D, S2A). However, MuSiC found a small but implausible renal tubular signal in most bulk transcriptomes, which was mostly absent from cellular signal analysis. This difference was even more pronounced for bulk transcriptomes where an adequate reference was not available (Fig. 1E, S2B). Here, MuSiC identified a substantial contribution from renal

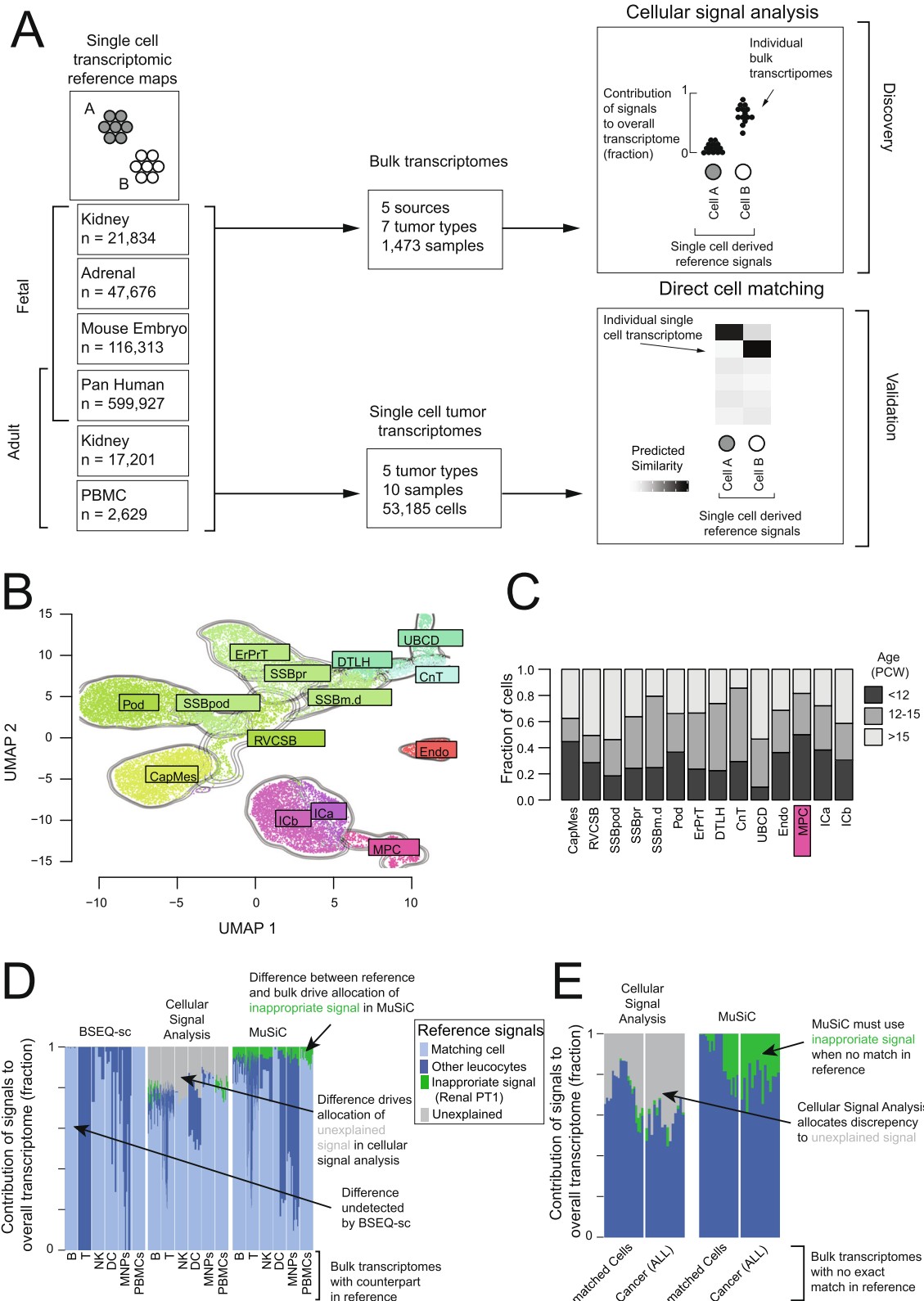

proximal tubular cells to pre-B cell leukemia, while BSeq-SC was unable to differentiate pre-B cell leukemia transcriptomes from normal mature B cells (Supplementary Fig. 2B). Cellular signal analysis identified pre-B cell leukemia as most similar to B cells, but with a substantial part of the signal unexplained by the given reference (Fig. 1E, Supplementary Fig. 2B).

As a further test, we applied all three methods to 766 ccRCC transcriptomes from The Cancer Genome Atlas[19] to assess whether the known cellular identity of these cancer cells could be identified. Cellular signal analysis best identified the signal of a specific proximal tubular cell population as the predominant cell signal in ccRCC bulk cancer transcriptomes (Supplementary Fig. 3).

**Fig. 1 Methodology overview and validation. A** Overview of methodology: Single cell reference atlases (left) define cellular signals. These are used to calculate the contribution of each cellular signal to bulk transcriptomes (top, Supplementary Data 1), where signal contributions are normalized to give a score between 0 and 1 for each bulk transcriptome, cellular signal pair (top right). These findings are validated by comparing the same cellular signals (left) to single cell tumor transcriptomes (bottom, Supplementary Table 1), where logistic regression generates a similarity score for each single cell transcriptome, cellular signal pair (bottom right). **B** Combined fetal kidney reference map: Contours and colors indicate the labeled cell type. *CapMes* Cap Mesenchyme, *RVCSB* Renal vesicle and comma-shaped body, *SSBpod* S-shaped body podocyte, *SSBpr* S-shaped body proximal tubules, *SSBm.d* S-shaped body medial and distal, *Pod* Podocytes, *ErPrT* Early proximal tubules, *DTLH* Distal tubule and loop of Henle, *UBCD* Ureteric Bud and collecting duct, *CnT* Connecting tubules, *Endo* Endothelium, *ICa* Interstitial cells a (smooth muscle), *ICb* Interstitial cells b (stromal), *MPC* Mesenchymal progenitor cells. **C** Age distribution of fetal kidney populations: Bar heights indicate fraction of cell type (as in **B**) by fetal age (color) in post conception weeks. **D** Benchmarking with match in provided reference: Comparison of two "bulk deconvolution" methods (BSEQ-sc and MuSiC) to cellular signal analysis, using bulk transcriptomes for which a good match exists in the reference single cell dataset. Bars height represent signal contributions from an immune cell and proximal tubular cell (PT1, included as a negative control) reference set in explaining bulk transcriptome from peripheral blood or flow sorted cells as indicated by the x-axis, bar color, and legend. "Matching cell" indicates a contribution from the expected signal (e.g., NK cell signal in NK bulk transcriptomes). See also Supplementary Fig. 2A. **E** Benchmarking with no match in provided reference: As in **D**, except bulk transcriptomes are flow sorted immune cells not in the reference (labeled "Unmatched cells") or pre-B cell acute lymphoblastic leukemias (labeled "Cancer ALL"). See also Supplementary Fig. 2B. Source data are available as a Source Data file.

To test the methods on a more traditional "deconvolution" metric, we applied cellular signal analysis and MuSiC to 100 pseudo-bulk transcriptomes constructed from the reference single cell data. We then estimated how accurately the known number of cells of each type that was used to construct the pseudo-bulk transcriptome could be recovered (Supplementary Fig. 2C, D). This comparison found that while cellular signal analysis had reasonable accuracy, MuSiC was consistently the best performing method (Supplementary Fig. 2C, D). This highlights that the cost of the flexibility built into cellular signal analysis in accommodating unexplained signals is lower accuracy in determining the cellular composition of bulk transcriptomes.

Taken together, these comparisons demonstrate the need for a bespoke approach to identify the main cellular signals in bulk transcriptomes where the reference data is incomplete. Cellular signal analysis quantifies the inadequacy of the reference through the allocation of "unexplained signal". Mathematically, this "unexplained signal" represents an intercept term, included to limit the assignment of spurious signals when a bulk transcriptome differs from all signals in the reference (see Methods). As the reference becomes progressively less suited to the bulk transcriptome being modeled, the "unexplained signal" contribution becomes steadily larger (Supplementary Fig. 2C).

**Childhood tumors, but not adult tumors, exhibit a fetal transcriptome.** For each tumor, we determined whether it exhibited a fetal or mature (i.e., post-natal) transcriptome, to guide the choice of reference in subsequent analyses. This analysis also enabled us to test two fundamental hypotheses about the differentiation state of tumors —that childhood tumors represent fetal cell types and that adult cancers, especially epithelial malignancies, dedifferentiate towards a fetal state. We define dedifferentiation to be the reversion of a mature cell to a fetal state, at the level of the whole transcriptome.

We calculated the immaturity by fitting each bulk transcriptome to a combined reference set composed of cellular signals from both mature and fetal kidney reference populations. The immaturity score was the fractional contribution of the developmental signals to the bulk transcriptome. Using this approach, we established a reference range of mature normal kidneys (Fig. 2A). We demonstrated the validity of this range by scoring fetal kidney transcriptomes which lay significantly outside the mature range ($p = 0.015$, Wilcoxon rank sum test).

We next calculated the same maturity score for individual tumors, which showed a clear signal of "fetalness" across all types of childhood kidney tumors (Fig. 2B, C). Although all childhood kidney tumors had a significant enrichment for developmental cellular signals, pretreated Wilms tumor had a significantly lower score than other childhood kidney tumors, including untreated Wilms. The comparison between treated and untreated Wilms suggests that chemotherapy reduces the developmental signal in Wilms tumor, a notion we explore in detail in a later section.

A significant developmental signal was absent from almost all adult tumors (Fig. 2B). This suggests that global "dedifferentiation" to a developmental state does not occur in adult kidney tumors. One obvious exception to the ubiquitous lack of a strong developmental signal in adult tumors ($p < 10^{-4}$, Wilcoxon rank sum test) was a cohort of lethal chromophobe RCC, classified previously as metabolically divergent due to their comparatively low expression of genes associated with the Krebs cycle, electron transport chain, and the AMPK pathway[19].

Motivated by this observation, we tested whether other clinical markers such as somatic genotype, morphology, or molecularly defined subgroup were predictive of immaturity score. We found that clear cell renal cell carcinomas with two independent somatic mutations in *PTEN* had a significantly higher immaturity score (Fig. 2D; t-test, FDR < 0.01). As with lethal chromophobe tumors, *PTEN* mutated ccRCCs conferred a far worse prognosis, with all samples belonging to the TCGA defined m3/ccB subgroup with the worst prognostic outcome of all groups[20]. Investigating further, we found an association between immaturity score and the m3 transcriptional subgroup (Fig. 2E; t-test, FDR < 0.01). No other clinical covariate had a statistically significant association with immaturity score at a 1% significance level (Supplementary Tables 2 and 3).

**Congenital mesoblastic nephroma resembles mesenchymal progenitor cells.** Congenital mesoblastic nephroma (CMN) is a renal tumor of infants that has low metastatic potential. There are two morphological subtypes of CMN, classical and cellular variants[21]. Cell signal analysis in CMN bulk transcriptomes ($n = 18$) revealed a uniform signal of mesenchymal progenitor cells across tumors (Fig. 3A, S4), irrespective of morphological subtype. Of note, these mesenchymal progenitor cells were characterized by expression of *NTRK3* and *EGFR* genes (Fig. 3B), the principal oncogenes that drive CMN through activating structural variants[22]. To verify that this signal was not a generic consequence of fibroblast-like cells, we repeated the analysis of bulk CMN transcriptomes using a developmental reference combined with mature fibroblasts. This comparison revealed the same match to mesenchymal progenitor cells, with a low contribution from mature fibroblasts (Fig. 3C).

To validate this mesenchymal stem cell signal in CMN, we subjected cells dissociated from a fresh CMN tumor specimen, to single cell mRNA sequencing using the Chromium 10x platform.

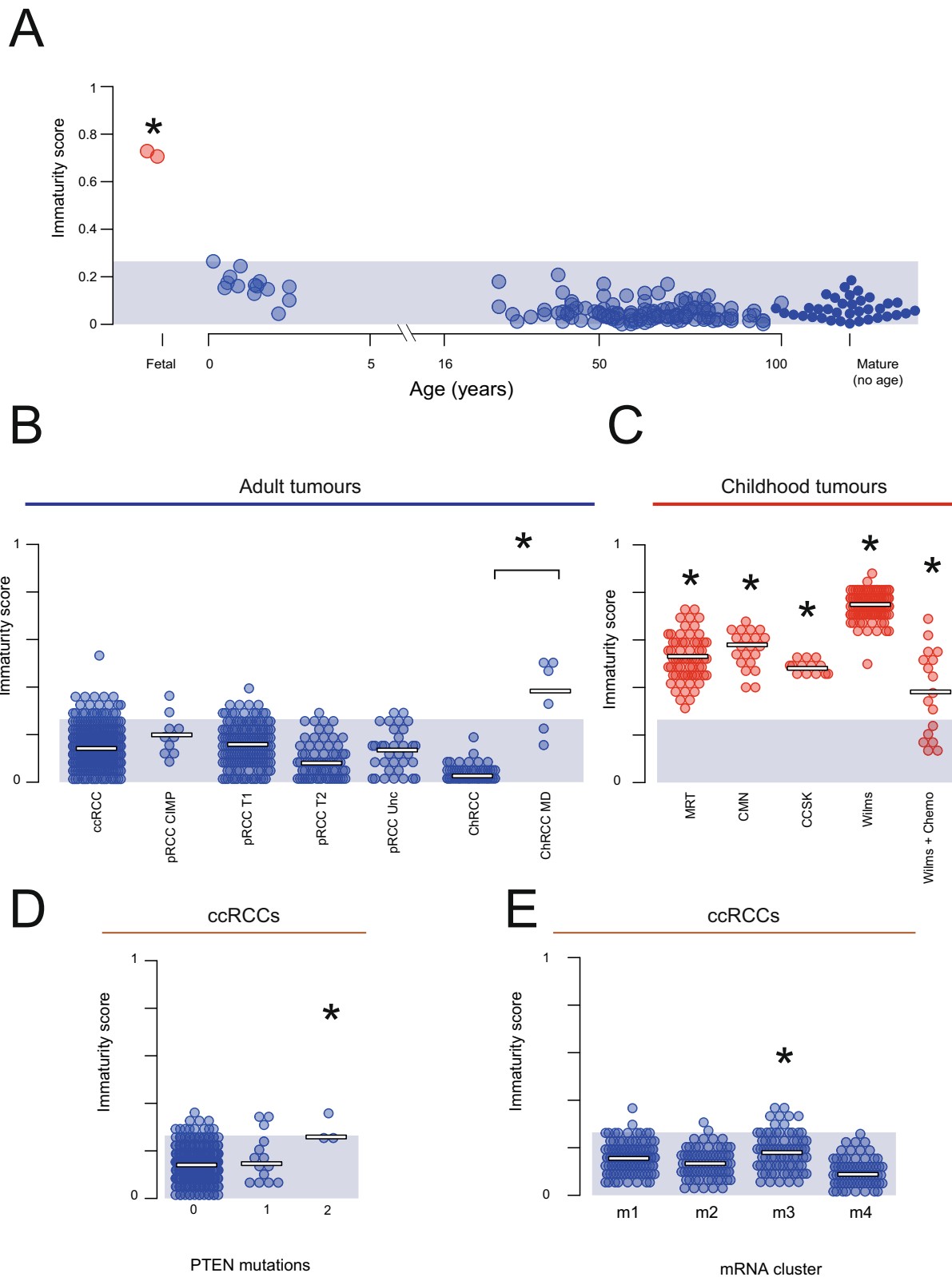

We annotated single cells based on literature derived marker genes (Fig. 3D, S5) and compared to single cell clusters of normal fetal kidneys using previously developed quantitative approaches[1]. This comparison revealed that CMN tumor cells matched the same mesenchymal progenitor cell population, validating the cell signal seen in bulk tumor tissue (Fig. 3E).

**Wilms tumor, clear cell sarcoma of the kidney and the effect of treatment**. Wilms tumor is the most common childhood kidney cancer and is thought to arise from aberrant cells of the developing nephron. Clear cell sarcoma of the kidney (CCSK) is a rare, at times aggressive childhood renal cancer that is treated as a high risk Wilms tumor in clinical practice[23]. We assessed the cellular

**Fig. 2 Immaturity score. A** Normal kidney: From 201 bulk transcriptomes from normal kidney an immaturity score was calculated by fitting each bulk transcriptome using a combined mature and fetal kidney cellular signal reference. The immaturity score is the total normalized signal contribution from fetal kidney in each bulk transcriptome (y-axis). The x-axis shows sample age, with unknown age on the right and fetal samples on the left in red. The shaded blue area indicates the range of maturity scores across all normal post-natal transcriptomes. The star indicates that fetal samples have maturity scores significantly higher than normal samples ($p = 0.015$, two-sided Wilcoxon rank sum test). **B** Adult renal tumors: Immaturity score (as in **A**) for 853 adult renal tumors, with normal immaturity score range shown by blue shading. The metabolically divergent subtype of Chromophobe renal cell carcinomas have a significantly different maturity score as indicated by the star ($p = 5.6 \times 10^{-6}$ two-sided Wilcoxon rank sum test). **C** Childhood renal tumors: Immaturity score (as in **A**) for 287 childhood renal tumors, with normal immaturity score range shown by blue shading. Each type of childhood tumor had a significantly different maturity score than post-natal normal tissue kidneys ($p < 2.2 \times 10^{-16}$ (MRT), $6.8 \times 10^{-14}$ (CMN), $1.8 \times 10^{-9}$ (CCSK), $< 2.2 \times 10^{-16}$ (Wilms), and $10^{-10}$ (Wilms + Chemo) two-sided Wilcoxon rank sum test). **D** ccRCCs by PTEN mutation status: Immaturity score for clear cell renal cell carcinomas as calculated in **A**, split by PTEN mutation status (0 = wild type, 1 = mono-allelic loss, 2 = bi-allelic loss). The star indicates that bi-allelic loss is a significant predictor of higher immaturity score ($p = 0.003$, two-sided t-test with multiple hypothesis correction). **E** ccRCCs by transcriptional group: Immaturity score for clear cell renal cell carcinomas as calculated in **A**, split by transcriptomic subgroups[20]. The star indicates that samples in m3 have a significantly lower immaturity score ($p = 2.9 \times 10^{-6}$, two-sided t-test with multiple hypothesis correction). Source data are available as a Source Data file.

signals in bulk transcriptomes from treatment-naive CCSK, high risk treatment-naive Wilms, and intermediate risk Wilms post chemotherapy. Cellular signal analysis revealed a largely uniform early nephron signal (cap mesenchyme, comma-shaped body, S-shaped body) in the treatment-naive Wilms cohort (Fig. 4A, S6). By comparison, the post-treatment cohort had a much-reduced contribution from the early nephron, instead containing a mixture of tubular, early nephron, and mesenchymal signals with a relatively high unexplained signal fraction (Fig. 4A, S6). Previous work utilizing single cell data from post-chemotherapy Wilms tumors identified the same lack of cap mesenchyme signal identified by our analysis of bulk transcriptomes[1]. The CCSK transcriptomes showed a mixture of mesenchymal and early nephron signals, with an extremely high unexplained signal fraction (Fig. 4A, S6).

To validate the cap mesenchyme signal in treatment-naive Wilms, we generated single cell mRNA transcriptomes from one fresh, treatment-naive sample. Annotation of this data revealed two proliferating populations (Fig. 4B, S7). Comparison to fetal kidney showed that one of these populations strongly matched the cap mesenchyme, validating its presence in treatment-naive Wilms tumor (Fig. 4C). The second population exhibited a strong match to mesenchymal progenitor cells (Fig. 4C).

To further investigate the origins of CCSK we generated single nuclear transcriptomes from 2 archival samples and single cell transcriptomes from one fresh sample (Fig. 4B, S8). In contrast to Wilms tumor, all CCSK tumor cells matched multiple mesenchymal and early nephron populations (Fig. 4C). Although the matching populations were consistent with the results of cell signal analysis on bulk CCSK transcriptomes (Fig. 4A), the match to multiple reference populations at the single cell level suggests that CCSK transcriptomes represent a transcriptional state that is intermediate between multiple mesenchymal populations in the developing kidney. To test the possibility that the true normal cell correlate for CCSKs was not in the fetal kidney, we next matched CCSK bulk transcriptomes against mature kidney, fetal adrenal, developing mouse, and the pan-tissue human cell landscape[24]. In each of these comparisons, the unexplained signal explained at least 50% of the CCSK bulk transcriptomes, a much higher fraction than any other tumor type (Fig. 4D). This unexplained signal fraction was comparable to the level obtained from a deliberately inappropriate comparison of flow sorted B cell bulk transcriptomes compared to the non-immune developing kidney (Fig. 4D). In aggregate, these data suggest that CCSKs represent transcriptionally grossly distorted renal mesenchymal cells.

**Malignant rhabdoid tumors exhibit signals of neural crest and early mesenchyme.** Malignant rhabdoid tumor (MRT) is an aggressive, often fatal childhood cancer, that typically affects the

kidney but may also occur at other sites. It is considered to be the extracranial counterpart of the CNS tumor, atypical teratoid/rhabdoid tumor (AT/RT). The principal, usually sole, driver event in MRT and AT/RT is biallelic inactivation of *SMARCB1*. In previous analyses of microRNA profiles, MRTs co-clustered with a range of tissues: neural crest derived tumors, cerebellum, and synovial sarcoma[25].

Assessing fetal renal single cell signals in 65 MRT bulk transcriptomes yielded a mesenchymal progenitor cell signal (Fig. 5A, S9). However, the nephron and unexplained signal fractions were also high, indicating that tumor cells only moderately resemble this reference population. To investigate further, we studied MRT single cell transcriptomes, derived from an MRT expanded by a primary organoid culture[26] (see Methods), from nuclear mRNA sequencing, and from fresh tissue MRT cells (Fig. 5B, S10). Comparison to our fetal kidney reference revealed that MRT cell transcriptomes did not show any consistent match (Fig. 5C). This may indicate that the mesenchymal progenitor cell signal obtained in bulk represents a signal of the broad embryological lineage of the tumor, rather than a cell type.

We therefore compared MRT cells against published reference cell populations of gastrulation embryos generated from mice[13], a developmental stage that is not accessible to study in humans. Although there were differences between and within samples, all produced a match to neural crest and/or early mesodermal/mesenchymal populations (Fig. 5C). To validate this early mesodermal signal, we performed immunohistochemistry for the presence of a protein specific to paraxial mesoderm, TWIST1. Consistent with its expression in a subset of cells by single cell mRNA sequencing, occasional MRT cells exhibited *TWIST1* staining, whilst no protein was detected in normal kidney (Fig. 5D, S11). Overall our data show that MRTs do not exclusively exhibit mRNA signals of either neural crest or mesenchyme cells. Instead, our findings point at a hybrid state of MRTs, representing mRNA features of both, neural crest and mesenchyme, suggesting that MRTs may come from early mesoderm or form along the differentiation trajectory of neural crest to mesenchyme.

**Adult tumors represent specific tubular cells.** As discussed above, our analyses confirmed a previous finding that the predominant single cell signal in the most common types of adult renal cancer, clear cell RCC (ccRCC) and papillary RCC (pRCC), was derived from a specific subtype of proximal tubular cells, termed PT1 cell (Supplementary Fig. 3)[1]. In addition, cell signal analysis also revealed some properties of the tumor microenvironment. We found a prominent vascular endothelial signal in ccRCCs (Fig. 6A, S12), but not in pRCCs. The downstream

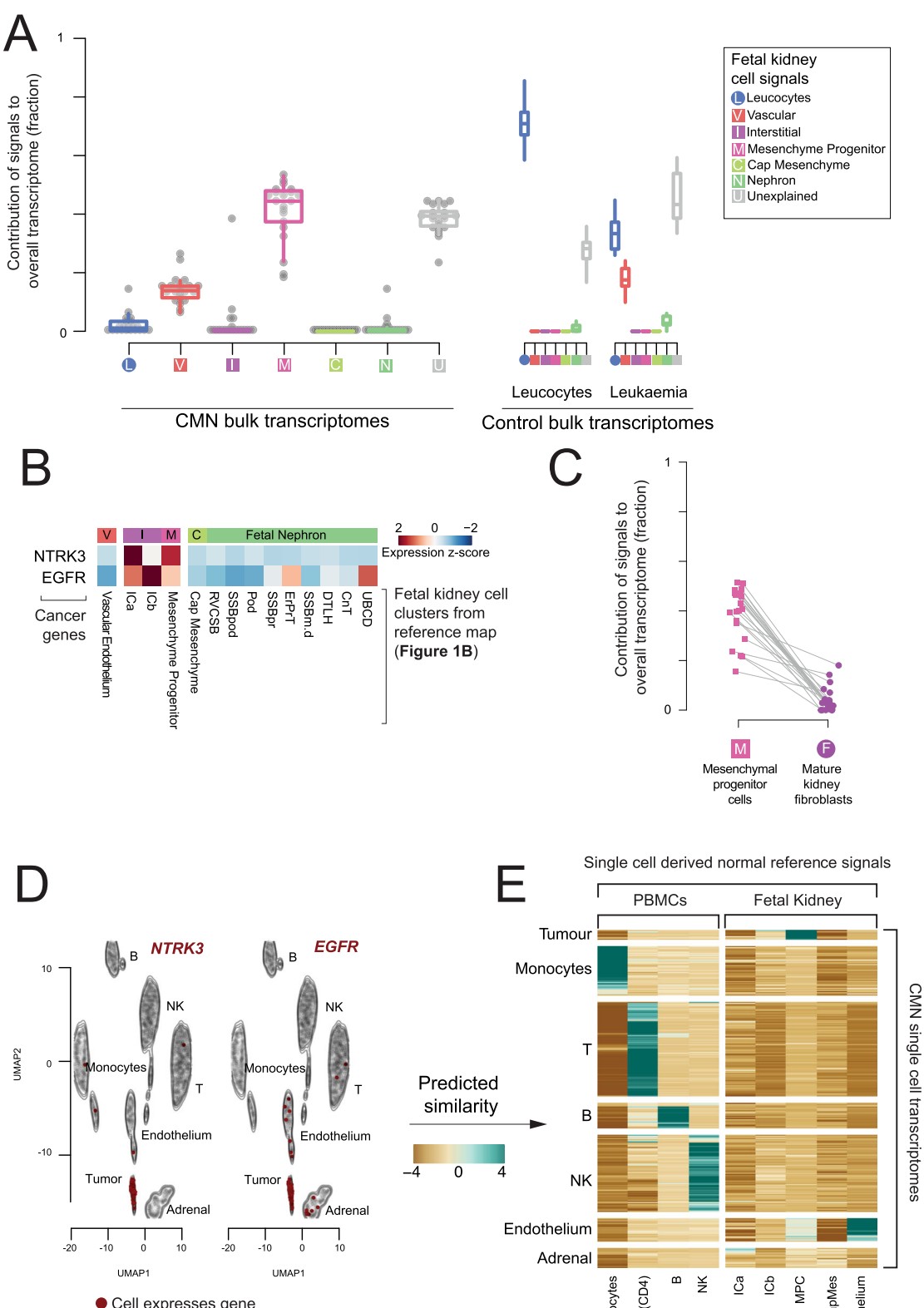

effects in RCC of inactivation of the von Hippel–Lindau gene and upregulation of vascular endothelial growth factors are well documented[27]. The prominent difference in the endothelial signal provides a read-out of this pathway, further explaining why anti-angiogenic treatments appear to be more effective in ccRCCs than in pRCCs[28].

Continuing our investigation of the tumor microenvironment, we observed mast cells to be over-represented in single cell data derived from pRCCs (Fig. 6B). Performing cellular signal analysis revealed a high contribution of mast cell signal in a subset of tumors, significantly enriched for type 1 pRCC tumors ($p < 1e$-4, Wilcoxon rank-sum test; Fig. 6C). This finding was further

**Fig. 3 Congenital mesoblastic nephromas. A** Composition of bulk CMNs: The relative contribution of single cell derived signals from fetal kidney in explaining the bulk transcriptomes of 18 congenital mesoblastic nephromas (CMNs) along with control leucocyte and ALL populations. The relative contribution of each signal to each bulk RNA-seq sample is shown by the y-axis. Each signal/sample combination is represented by a single point and boxplots shows the distribution with median (middle line), 1st and 3rd quartiles (box limits) and 1.5 times the inter-quartile range (whiskers). Each signal type is abbreviated and colored as per the legend, with squares for fetal and circles for mature. CMN samples are shown on the left and control samples on the right, where "Leukocytes" are bulk transcriptomes from flow sorted leukocytes and "Leukemia" represent B-precursor acute lymphoblastic leukemia. **B** Expression of CMN cancer genes in fetal kidney: Expression of CMN driver genes (rows) in reference fetal kidney single cell RNA-seq populations (columns), scaled to mean 0 and standard deviation 1 in each row (i.e., z-transformed). **C** Comparing mesenchymal progenitor cell signals to mature fibroblasts: All 18 CMN bulk transcriptomes were analyzed using a reference signal set including both fetal kidney cells and the fibroblasts from mature kidney. This figure shows the comparison of their inferred contribution to each transcriptome for each sample (y-axis), with lines joining points representing the same sample. **D** Expression of CMN marker genes: tSNE map of 4,416 single cell transcriptomes from a CMN biopsy, where contours indicate clusters of cells of the type labeled. Cells positive for *NTRK3* (left) and *EGFR* (right) are colored red. *B* B cell, *T* T cell, *DC* dendritic cell, *NK* NK cell, *NKT* NKT cell. **E** Comparison of single cell CMN to fetal kidney: Comparison of clusters of cells from **D**. (rows) with fetal kidney and leucocyte reference populations (columns). For each CMN cluster/reference population pair a log-similarity score was calculated using logistic regression (see Methods). Positive log-similarity scores represent a high probability of similarity between the reference and test cluster. Source data are available as a Source Data file.

validated by single molecule fluorescence in-situ hybridization (smFISH), which found a higher fraction of mast cells a type 1 pRCC sample, than type 2 or ccRCC (Fig. 6D, Supplementary Table 4).

Previous analyses of chromophobe cell renal cell carcinoma (ChRCC) have shown that ChRCC exhibit expression profiles of collecting duct cells[29]. Controversy exists as to whether the normal cell correlate of ChRCC is the type A or type B intercalated cells[30]. This is in part due to ChRCC retaining expression of both canonical markers of intercalated cells, SLC4A1 and SLC26A4 respectively (Supplementary Fig. 13). Using cell signal analysis, which considers the entire transcriptomes of type A and type B cells, rather than just two markers, revealed a uniform type A signal across all chromophobe tumors (Fig. 6E, S14), bar the lethal variant of so-called metabolically divergent tumors (Fig. 2B, S14). The proliferation and active remodeling of type A cells has been demonstrated under conditions of systemic acidosis[31], lending further credence to their possible status as the cell of origin for ChRCCs.

**Single cell signals provide diagnostic clues**. An overarching finding of our study was that each tumor type possesses a particular pattern of cellular signals that were uniform in, and specific to, bulk transcriptomes from individual tumor types. Accordingly, cellular signal assessment of bulk transcriptomes may provide sensitive and specific diagnostic clues. To test this proposition, we assessed how accurately the tumor type of each sample in our data could be determined based only on its cellular signals. We found that the prevalence of the most common cellular signal for each type could be used to infer the tumor type of each bulk transcriptome (Fig. 7A, B, S15). As further validation, we applied this approach to an independent cohort of Wilms tumors. All were correctly identified as childhood tumors and had cellular signals consistent with Wilms tumor (Supplementary Fig. 16).

We next examined cellular signals in the bulk transcriptome of a histologically undefinable metastatic primary renal tumor from an 11-year-old boy. Following resection, the tumor was examined histologically, both locally and by international reference renal pathologists (Fig. 7C). A definitive diagnosis could not be reached although an adult type renal cell carcinoma was favored. Nevertheless, the child was treated as a Wilms-like tumor, with cytotoxic chemotherapy and radiotherapy, following nephrectomy. He remains in complete remission two years following diagnosis, thus retrospectively suggesting a diagnosis of a Wilms-like tumor, as adult type kidney carcinomas do not respond to cytotoxic treatment.

We performed bulk mRNA sequencing on tumor specimens from this patient. Assessment of mRNA signals in bulk tissue suggested that the tumor exhibited a fetal transcriptome with cellular signals consistent with a Wilms-like tumor (Fig. 7D, E). The transcriptional diagnosis of a Wilms-like tumor was further substantiated by analyses of whole genome sequences. The tumor harbored classical somatic changes of Wilms, namely canonical *CTNNB1* and *KRAS* hotspot mutations and uniparental disomy of 11p (Supplementary Fig. 17). By comparison, when we assessed single cell signals of an adult-type ccRCC that developed in a 15 year old adolescent, we found an overall mature transcriptome. Furthermore, the tumor exhibited the PT1 signal of ccRCC as well as a stark vascular endothelial signal typical of ccRCC (Fig. 7F, G).

## Discussion

We have determined normal cell signals in the major types of human renal tumors. This has enabled us to replace the approximate notion of the "fetalness" of childhood renal tumors with quantitative transcriptional evidence that the entire spectrum of pediatric renal tumors represent an aberrant developmental state. At the same time, our analyses question the suggestion that adult, epithelial-derived kidney cancers revert to a fetal state at the whole transcriptome level (i.e., "dedifferentiate"). Importantly, when we found transcriptional evidence of dedifferentiation in adult tumors, it conferred a dismal prognosis. Furthermore, among childhood tumors we found examples of cell signals representing differentiation trajectories, such as the neural crest to mesenchyme conversion in MRT, validating our recent finding based on phylogenetic and differentiation studies[32]. By contrast, the different types of adult tumors resembled specific renal tubular cells.

A central question that our findings raise is whether mRNA signals point to the cell of origin of tumors. When the similarity between mRNA signals and specific cell types was high, as found in most tumor types, this may be a plausible proposition. For example, in CMN, which typically occurs within the first weeks of life, our analysis identified an early mesenchymal progenitor cell population, characterized by the disease-defining oncogenes of CMN, as the likely cell of origin of CMN. In some tumors, transformation may entirely distort and obliterate gene expression profiles of the cell of origin. We found CCSK transcriptomes to represent such an extreme modification of the transcriptome of the developing kidney.

A further finding of our study was that within each category, the majority of tumors exhibited remarkably uniform cellular signals. That is, despite a high diversity in clinical outcome,

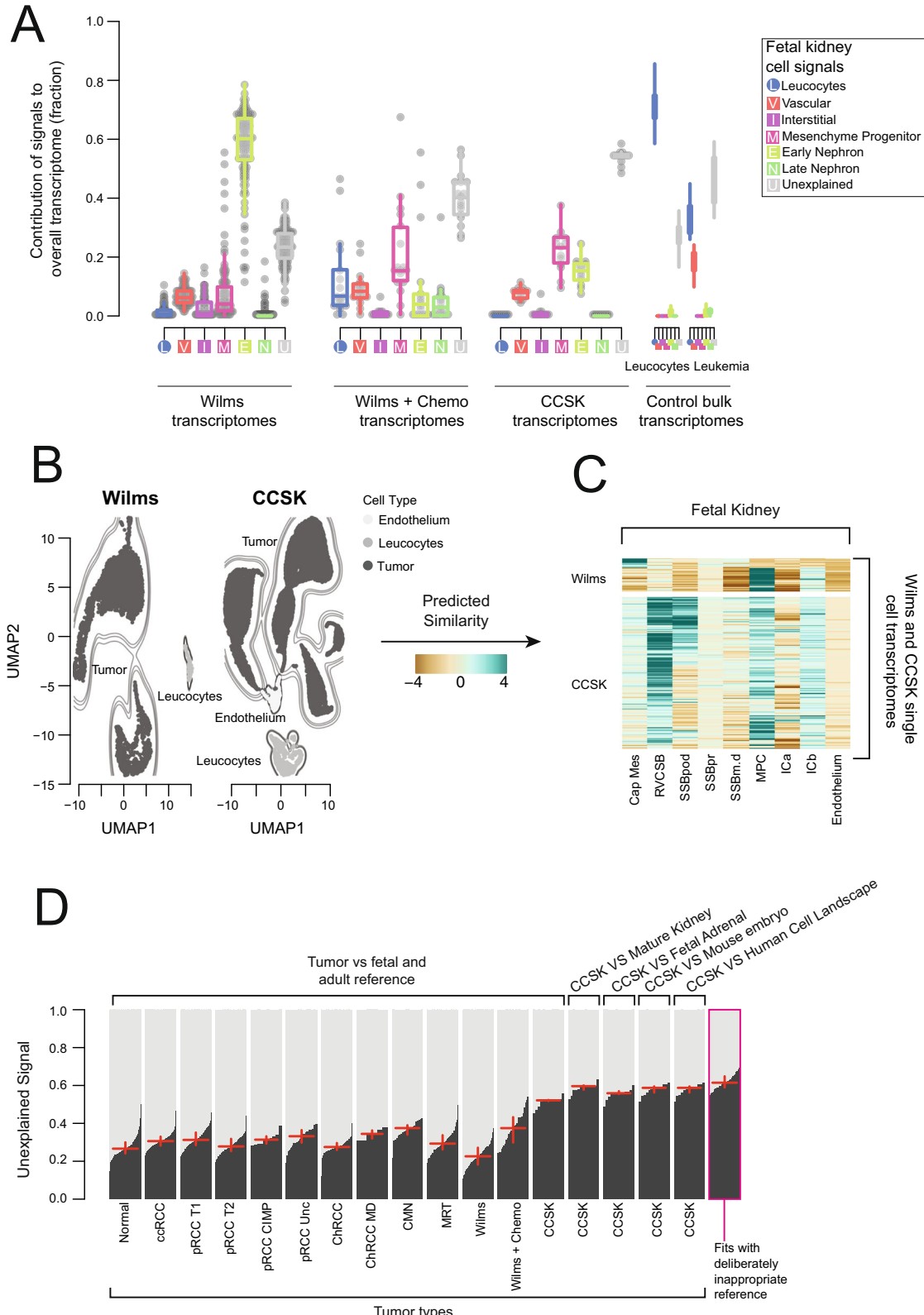

tumors of the same type almost universally had the same dominant cellular signal (Fig. 7A, B). This indicates that there are overarching transcriptional features, beyond individual gene markers, that unite tumor entities despite underlying intra- and inter-tumor genetic heterogeneity. Therefore, cellular signals of renal tumors may lend themselves as diagnostic adjuncts, as illustrated here by our ability to resolve the identity of a

histologically unclassifiable childhood tumor. Moreover, the cellular transcriptome itself may represent a therapeutic target that transcends individual patients, if we had tools available to manipulate transcription in a predictable manner. This may be a particularly attractive approach for targeting transcriptional states of fetal cells retained in childhood cancer that are absent from normal post-natal tissues.

**Fig. 4 Wilms tumor and clear cell sarcoma of the kidney. A** Bulk Wilms tumor and CCSK compared to fetal kidney: The relative contribution of single cell derived signals from fetal kidney in explaining the bulk transcriptomes of 137 nephroblastomas (Wilms tumors) and 13 CCSKs along with control populations. The relative contribution of each signal to each bulk RNA-seq sample is shown by the y-axis. Each signal/sample combination is represented by a single point and boxplots shows the distribution with median (middle line), 1st and 3rd quartiles (box limits) and 1.5 times the inter-quartile range (whiskers). Each signal type is abbreviated and colored as per the legend, with squares for fetal and circles for mature. Wilms/CCSK samples are shown on the left and control samples on the right, where "Leukocytes" are bulk transcriptomes from flow sorted leukocytes and "Leukemia" represent B-precursor acute lymphoblastic leukemia. **B** UMAP of CCSK and Wilms single cell transcriptomes: Points represents cell transcriptomes from Wilms tumor (left) or CCSK (right), with shading, contours, and labels indicate cell type. **C** Comparison of CCSK and Wilms transcriptomes to reference signals: Similarity of transcriptomes from **B** (rows) to fetal kidney reference signals (columns), where color indicates logit similarity. **D** Comparison of unexplained signal contribution to CCSKs and other tumor types: For each group of samples, the unexplained signal is calculated using the reference set of signals given at the top (e.g., fetal kidney). The unexplained signal fractions are shown by black bars, sorted in increasing order, with the red horizontal line showing the median value and the vertical line the range between the 25th and 75th percentiles. CCSK samples were fitted using 5 different reference sets (fetal and mature kidney, mature kidney only, fetal adrenal, mouse embryo, and the pan-tissue human cell landscape). The final group on the right, represents samples fitted using inappropriate references. This population serves as a calibration of the expected level of unexplained signal when the bulk transcriptome is not explained by any of the provided reference signals. Source data are available as a Source Data file.

Overall our findings attach specific cell labels to human renal tumors that are underpinned by quantitative molecular data obtained from single cell mRNA sequences, independent of the interpretation of marker genes. As reference data from single cell transcriptomes expand through efforts such as the *Human Cell Atlas*, it will be feasible to annotate existing large repositories of tumor bulk transcriptomes, to derive a cellular transcriptional definition of human cancer.

## Methods

**Ethics statement**. Human kidney and tumor tissues were collected through studies approved by UK NHS research ethics committees. Patients or guardians provided informed written consent for participation in this study as stipulated by the study protocols. These studies have the following references: NHS National Research Ethics Service reference 03/018 (DIAMOND study; adult kidney tissues); NHS National Research Ethics Service reference 16/EE/0394 (pediatric tissues); NHS National Research Ethics Service reference 96/085 (fetal tissues). Additional fetal tissue was provided by the Joint MRC/Wellcome Trust-funded (grant # 099175/Z/12/Z) Human Developmental BiologyResource (HBDR, http://www.hdbr.org; (10)), with appropriate maternal written consent and approval from the Newcastle and North Tyneside NHS Health Authority Joint Ethics Committee. HDBR is regulated by the UK Human Tissue Authority (HTA;www.hta.gov.uk) and operates in accordance with the relevant HTA Codes of Practice. Fetal tissues from both sources were obtained from terminations and ranged from 7 to 18 post conception weeks (Supplementary Table 1). Organoids were generated from human tissue as approved by the medical ethics committee of the Erasmus Medical Center (Rotterdam, the Netherlands).

### Tissue processing and data generation
*10X single cell sequencing of fresh tissue and bulk sequencing of DNA/RNA*. Fresh tissues were processed to generate single suspensions for processing on the Chromium 10X controller (V2/3 3′ chemistry), as previously described[1]. The MRT and normal kidney tissue organoids were derived and maintained, as previously described[26]. Libraries were produced according to the manufacturer's instructions and sequenced on an Illumina HiSeq4000 device. Sequencing of bulk RNA and DNA was performed, as previously described[1].

*Cell-Seq2 experiments*. Following resection, a random piece was selected from viable tumor tissue, minced, and viably frozen. On the day of the sorting, the sample was thawed and dissociated into a single-cell suspension in AdDF + ++ (Advanced DMEM/F12 containing 1× Glutamax, 10 mM HEPES and antibiotics) containing Collagenase 1a (1 mg/mL, Sigma, C9407) and DNase (0.25 µg/mL, Stemcell), supplemented with Rho-kinase inhibitor Y-27632 (10 µM, Abmole). The samples were digested on an orbital shaker for 30 min at 37 °C. The suspension was washed first with AdDF + ++ and next with MACS buffer (PBS pH 7.2 + 2 mM EDTA + 0.5% Bovine Serum Albumine), followed each time by centrifugation at 300 × g. Viable single cells were sorted based on forward/side scatter properties and DAPI/DRAQ5 staining using FACS (MoFlo Astrios EQ, Beckman Coulter) into 384-well plates (Biorad) containing 10 µl mineral oil (Sigma) and 50 nl of RT primers.

*10X single nuclei sequencing*. Single nuclei were isolated from frozen tissue using a glass dounce homogeniser. Samples were homogenised in buffer A (Sucrose 0.25 M, BSA 10 mg/ml, MgCl$_2$0.005 M, protease inhibitors and RNAse inhibitors RNAseIn—0.12 U/ul and Superasin 0.06 U/ul), using ~25 strokes with the "loose" pestle and ~20 strokes with the "tight" pestle. Nuclei were cleaned up using a 30%

Percol gradient and resuspended in buffer B (Sucrose 0.32 M, BSA 10 mg/ml, CaCl2 3 mM, MgAc2 2 mM, EDTA 0.1 mM, Tris-HCl 10 mM, DTT 1 mM in the presence of protease and RNAse inhibitors as in buffer A).

Nuclei were mixed 1:1 with Tryphan blue and counted using a disposable haemocytometer, then diluted to the appropriate concentration. Nuclei were loaded on to the 10X Chromium controller as per the Chromium Single Cell 3′ Reagent Kits v3 User Guide, targeting to recover 5000 nuclei. Post GEM-RT cleanup, cDNA amplification, and 3′ gene expression library construction were carried out according to the user guide. The resulting libraries were sequenced on the Novaseq platform.

*Immunohistochemistry of MRT tissue*. Immunohistochemistry was performed on 3–4 µm sections of tissue fixed in 4% paraformaldehyde, dehydrated, and embedded in paraffin according to standard protocols. Sections were subjected to H&E and immunohistochemical staining using antibodies for INI-1 (BD Transduction Laboratories, 612111, 1:400) or TWIST (Abcam, ab50581, 1:500). Counterstaining was performed using Mayer's Hematoxylin (1:3 dilution). The Leica DMi8 microscope was used for imaging.

*RNAscope smFISH and immunohistochemistry*. FFPE tissue sections of 5 µm thickness were processed using a Leica BOND RX to automate staining with the RNAscope Multiplex Fluorescent Reagent Kit v2 Assay and RNAscope 4-plex Ancillary Kit for Multiplex Fluorescent Reagent Kit v2 (Advanced Cell Diagnostics, Bio-Techne) in combination with immunohistochemistry (IHC) for PECAM1[33]. Owing to intense tissue autofluorescence, samples were treated with a photo-bleaching procedure based upon that of Lin et al.[34]. It was observed that photo-bleaching prior to RNAscope probe hybridisation adversely affected smFISH staining, presumably due to loss of RNA integrity in the alkaline solution. Therefore, photobleaching was conducted following RNAscope probe and tree amplification reagents (AMP1/2/3) but before channel-specific HRP reagents and fluorophores. Initial automated processing included baking at 60 °C for 30 min and dewaxing, as well as heat-induced epitope retrieval at 95 °C for 15 min in buffer ER2 and digestion with Protease III for 15 min. Following RNAscope probe and AMP hybridisation according to the manufacturer's instructions, slides were briefly rinsed in PBS and then subjected to photobleaching. Slides were incubated horizontally in 4.5% hydrogen peroxide, 24 mM sodium hydroxide in PBS in a Nunc Square BioAssay Dish atop a white lightbox for 30 min. Slides were thoroughly rinsed with sterile deionised water and Leica BOND wash before RNAscope staining was resumed with the sequential development of three probe channels using tyramide signal amplification with Opal 570, Opal 650 (both Akoya Biosciences), and TSA-biotin (TSA Plus Biotin Kit, Perkin Elmer) and streptavidin-conjugated Atto 425 (Sigma Aldrich). Finally, IHC was carried out, beginning with a blocking step of 1 h in Primary Antibody Diluent (Leica), followed by rabbit anti-PECAM1 (Abcam ab28364) at 1:600 at room temperature for 2 h, and then HRP goat anti-rabbit IgG (Thermo G21234) at 1:1000 at room temperature for 1 h. Both antibodies were diluted in Primary Antibody Diluent. IHC signal was developed using Opal 520 (Akoya).

Stained sections were imaged with a Perkin Elmer Opera Phenix High-Content Screening System, in confocal mode with 1 µm z-step size, using a 20× water-immersion objective (NA 0.16, 0.299 µm/pixel). Channels: DAPI (excitation 375 nm, emission 435–480 nm), Atto 425 (ex. 425 nm, em. 463–501 nm), Opal 520 (ex. 488 nm, em. 500–550 nm), Opal 570 (ex. 561 nm, em. 570–630 nm), Opal 650 (ex. 640 nm, em. 650–760 nm).

### Basic data processing and quality control
*Mapping of DNA reads*. DNA sequencing reads were aligned to the GRCh 37d5 reference genome using the Burrows–Wheeler transform (BWA-MEM)[35]. Sequencing depth at each base was assessed using Bedtools coverage v2.24.0.

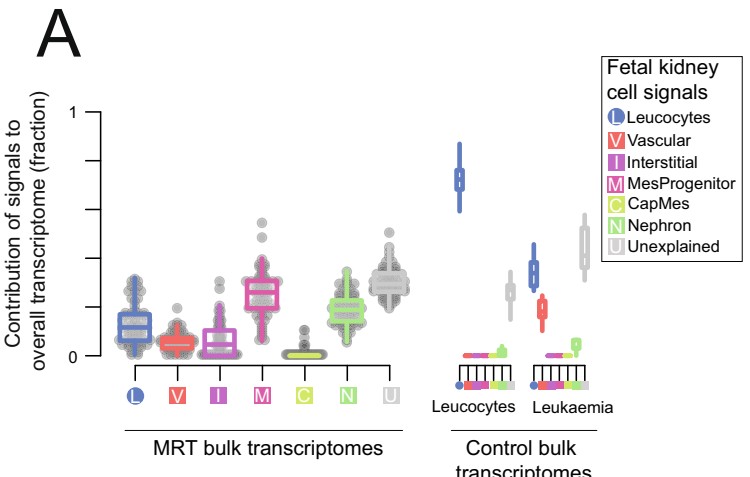

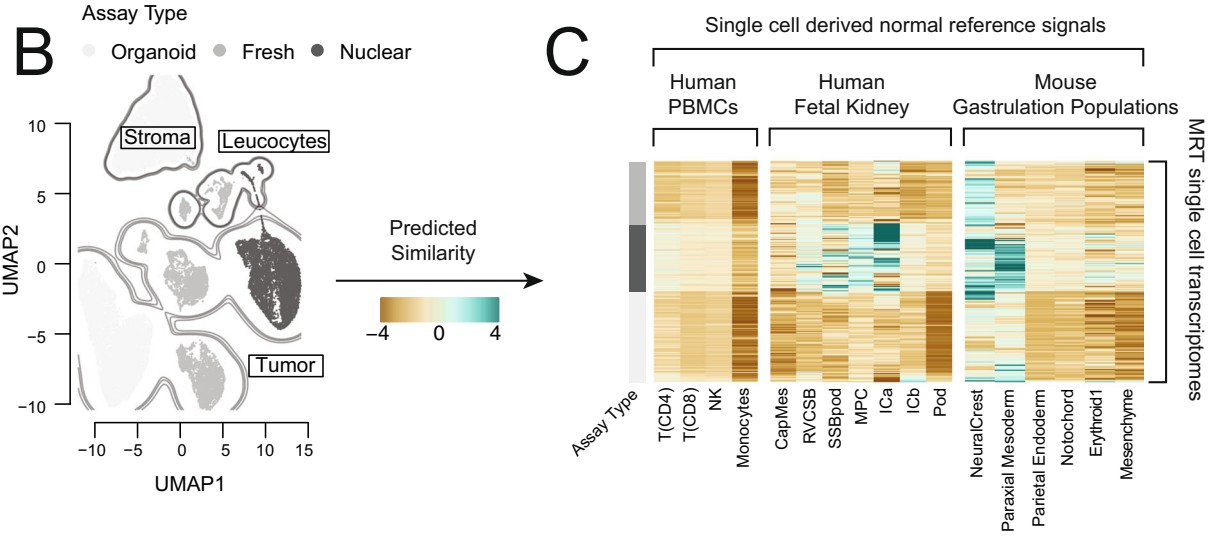

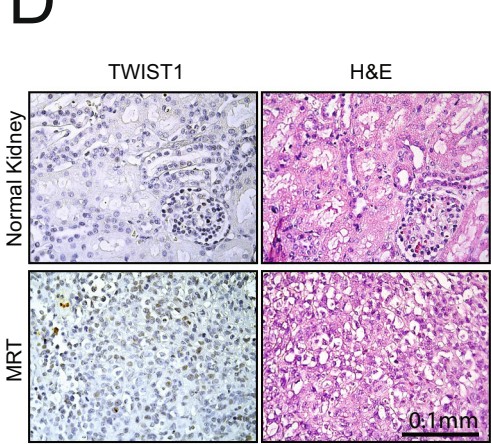

*Substitution calling.* Single base somatic substitutions were called using an in-house version of CaVEMan v1.11.2 (Cancer Variants through Expectation Maximization)[36]. CaVEMan compares sequencing reads from tumor and matched normal samples and uses a naive Bayesian model and expectation-maximization approach to calculate the probability of a somatic variant at each base (https://github.com/cancerit/CaVEMan). Small insertions and deletions (indels) were called using an in-house version of Pindel (v2.2.2; github.com/cancerit/cgpPindel). Post-

processing filters required that the following criteria were met to call a somatic substitution:

1. At least a third of the reads calling the variant had a base quality of 25 or higher.
2. If coverage of the mutant allele was less than 8, at least one mutant allele was detected in the first 2/3 of the read.

**Fig. 5 Malignant rhabdoid tumors. A** Bulk MRTs compared to fetal kidney: The relative contribution of single cell derived signals from fetal kidney in explaining the bulk transcriptomes of 65 malignant rhabdoid tumors (MRTs) along with control populations. The relative contribution of each signal to each bulk RNA-seq sample is shown by the y-axis. Each signal/sample combination is represented by a single point and boxplots shows the distribution with median (middle line), 1st and 3rd quartiles (box limits) and 1.5 times the inter-quartile range (whiskers). Each signal type is abbreviated and colored as per the legend, with squares for fetal and circles for mature. MRT samples are shown on the left and control samples on the right, where "Leukocytes" are bulk transcriptomes from flow sorted leukocytes and "Leukemia" represent B-precursor acute lymphoblastic leukemia. **B** UMAP of single cell MRT transcriptomes: Each dot represents a single transcriptome from either tumor/tubular derived organoid cells (white), fresh tissue MRTs cells (gray) or archival MRT nuclei (black). Contours indicate tumor cells, stroma, and leucocytes as labeled. **C** Log similarity of single cell MRT cells to fetal kidney and developing mouse: Comparison of the transcriptomes in **B** to cellular signals defined from single cell reference transcriptomes. The reference population is indicated on the x-axis and the gray bar on the left indicates the technology each cell was derived from. Each row corresponds to a single transcriptome from **B**. The color scheme encodes the logit similarity score for each cell against each reference population (see Methods). **D** Immunohistochemistry of *TWIST1* in MRT and normal kidney: Staining of a region of normal kidney and MRT tissue for *TWIST1*. The MRT image shows a part of the tissue selected for being *TWIST1* positive, there were large sections of tumor tissue that were also *TWIST1* negative. All normal kidney tissue was *TWIST1* negative. This experiment was repeated 3 times and the scale bar (bottom-right) indicates 0.1 mm. Source data are available as a Source Data file.

3. Less than 5% of the mutant alleles with base quality ≥ 15 were found in the matched normal.
4. Bidirectional reads reporting the mutant allele.
5. Not all mutant alleles reported in the second half of the read.
6. Mean mapping quality of the mutant allele reads was ≥ 21.
7. Mutation does not fall in a simple repeat or centromeric region.
8. Position does not fall within a germline insertion or deletion.
9. Variant is not reported by ≥ 3 reads in more than one percent of samples in a panel of approximately 400 unmatched normal samples.
10. A minimum 2 reads in each direction reporting the mutant allele.
11. At least 10-fold coverage at the mutant allele locus.
12. Minimum variant allele fraction 5%.
13. No insertion or deletion called within a read length (150 bp) of the putative substitution.
14. No soft-clipped reads reporting the mutant allele.
15. Median BWA alignment score of the reads reporting the mutant allele ≥140.
16. The following variants were flagged for additional inspection for potential artefacts, germline contamination or index-jumping event:
17. Any mutant allele reported within 150 bp of another variant.
18. Mutant allele reported in >1% of the matched normal reads.

*Copy number detection in bulk DNA.* The ascatNGS algorithm (v4.0.1)[37] was used to estimate tumor purity and ploidy and to construct copy number profiles prior to running the Battenberg algorithm (v2.2.5) (github.com/cancerit/cgpBattenberg) to allow for tumor subclonality.

*Bulk RNA mapping and quantification.* Where possible, we have processed all bulk RNA-seq data using the exact same pipeline as the recount2 project[38]. That is, we used RAIL-RNA to produce counts of bases aligned to each gene in each sample[39]. Counts were then converted to fragments aligned to genes by dividing counts by the average fragment length for the sample. This approach allowed us to combine our in-house data with any dataset processed by the recount2 project, in particular the TCGA and GTEX projects.

The length for each gene was calculated as the sum of unique exonic bases for all transcripts associated with each gene. We used the gencode v25 GTF annotation[40] and GRCh38 human reference genome.

In order to run the recount2 pipeline, we required access to the sequencer output (BAM files or fastq). In some cases, we only had access to processed data, either in the form of raw fragment counts, or transcripts per million (TPM). TPMs were converted to fragment counts by multiplying by 1 million, rounding to an integer and assigning each gene an effective length of 1.

Where TPM values were needed for direct comparison of gene expression, we calculated TPM values from fragment counts by dividing by gene length, then normalizing the counts/bp by forcing them to sum to 1,000,000 across all genes in a sample.

*Single cell RNA mapping, quantification, quality control, and normalization.* Single cell RNA-seq data were quantified using the 10X software package cellranger (version 2.0.2 for V2 chemistry, 3.0.2 for V3 chemistry) to map sequencing data to version 2.1.0 of the build of the GRCh38 reference genome supplied by 10X.

Data were normalized for sequencing depth by dividing by the total number of UMIs in each cell and then transformed to a log scale for each cell using the Seurat version 3.1.4[41] NormalizeData function. That is, the transformed data, y, is given by:

$$y_{gc} = \log\left(1 + F\frac{x_{gc}}{\sum_g x_{gc}}\right)$$

where x is the UMI count matrix with g indexing gene and c indexing the cell. F is the Seurat "scale.factor" parameter (which we left at the default value of 10,000).

Doublets were determined using scrublet[42] and ambient RNA contamination was removed with SoupX[43]. To filter lower quality cells, we performed high resolution clustering (Seurat graph-based clustering with resolution = 10) and filtered any cell which:

1. had greater than 5% expression originating from mitochondrial genes
2. was marked as a doublet
3. expressed fewer than 500 distinct transcripts
4. or belonged to a cluster where greater than 50% of cells failed one of 1–3.

The rationale behind this approach was to conservatively remove cells with a very similar transcriptome to cells which have failed QC.

To prevent similarity to reference maps (e.g., fetal kidney) being driven by cell cycle state, we also removed any cell with evidence of being in S or G2M phase. We determined the cell cycle phase by scoring each cell based on panel of genes specific to each phase using the Seurat CellCycleScoring function. We also removed all leucocytes from each tissues reference map.

### Analysis of processed data

*Derivation of color scheme.* In deriving a color scheme to represent the different types of cellular signal used in this paper, we started by designating a series of hue ranges to represent each tissue type. These hue ranges were then further sub-divided to represent more specific cell types. To separate fetal and mature versions of the same cell type, we used different values of the "value" parameter in hue, saturation, value color space to represent fetal (0.9) and mature (0.7) cell signals. Finally, we set the saturation value to 0.6 by default and allowed this to vary as necessary to emphasize differences between cell types with otherwise similar colors. This color scheme is summarized in Supplementary Fig. 18.

We also constructed a color scheme for each sample type in this study. We used light/pastel colors to represent non-tumor or control samples and solid colors for tumors. We used the same color to represent Neuroblastoma and ChRCC tumors as they were never referenced in the same plot. This color scheme is summarized in Supplementary Fig. 19.

*Dimension reduction and cluster generation of single cell RNA data.* Following normalization, we identified genes with high variability using the Seurat Find-VariableGenes function. This function calculates the mean expression and dispersion for each gene, then groups genes into bins (of size 20) by their mean expression and identifies any gene for which the z-score calculated from the dispersion exceeds some cut-off. We used the default cut-off of z = 1 and mean expression in the range 0.1 to 8.

The normalized data were scaled to have mean 0 and standard deviation 1 and principle component analysis was performed using the variable genes identified together with any gene that we identified as being potentially biologically interesting (regardless of its variability in the data).

We determined the optimal number of principle components (PCs) using molecular cross-validation (https://github.com/constantAmateur/MCVR)[44]. We used these to construct a two-dimensional representation of the data using either tSNE[45,46] or UMAP[47]. This representation was then used only to visualize the data.

Clusters were identified using the community identification algorithm as implemented in the Seurat "FindClusters" algorithm. We used the number of PCs determined above as input to this method and set the resolution parameter to 1. We chose this value of the resolution parameter as it produced a number of clusters that was large enough to capture most of the important biological variability but not so large as to make detailed manual scrutiny of each cluster impractical. All other parameters were set to the function defaults.

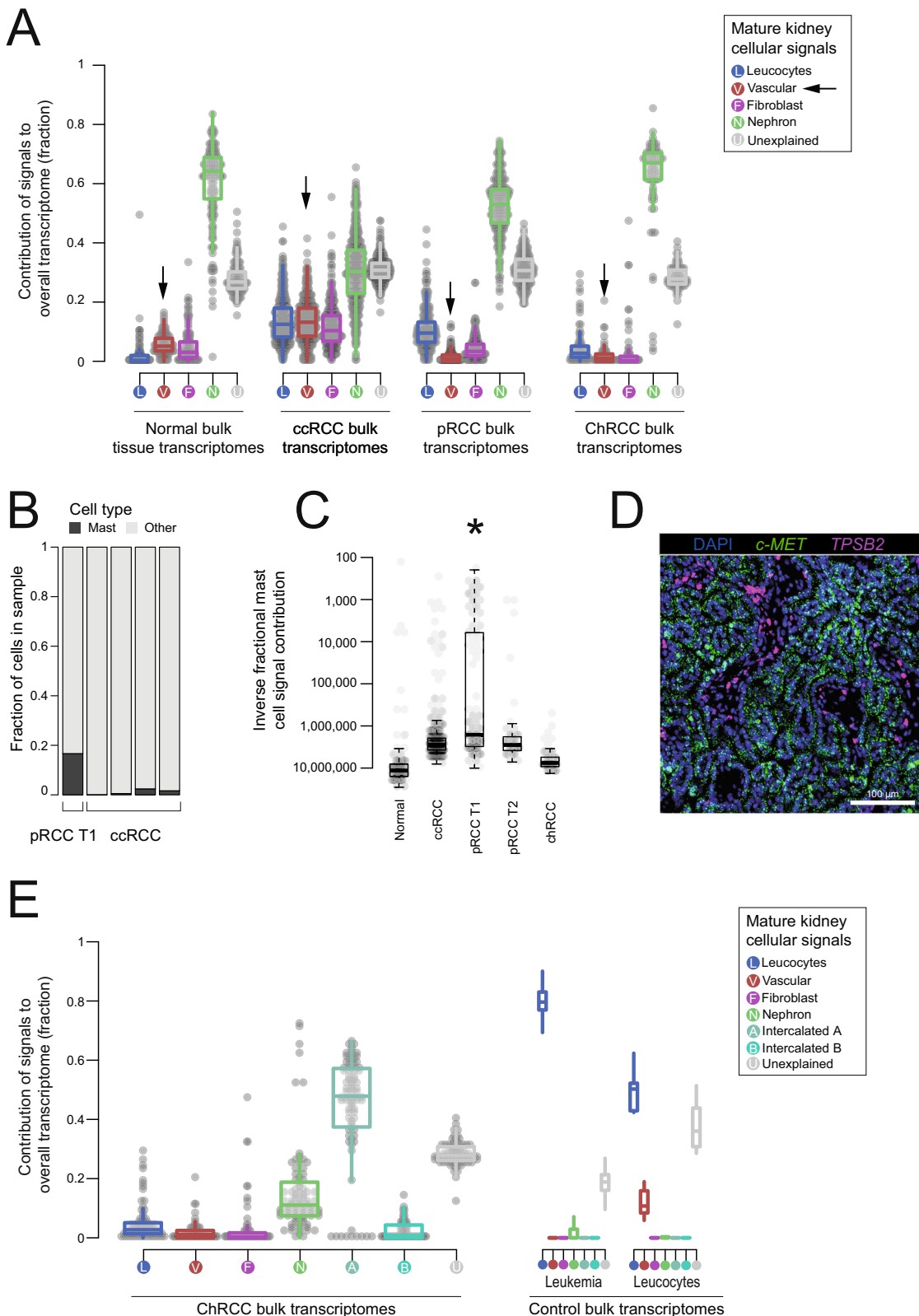

Annotation of fetal kidney single cell data. To create the fetal kidney reference, we combined the raw 10X output from previous studies[1,8] together with data from 4 additional fetal kidneys. The combined data were quantified and clustered as described above, with the exception that the clustering resolution parameter was set to 2 to obtain a more granular annotation.

To annotate these clusters, we used a previously published detailed annotation of the fetal kidney as a reference[8]. That is, we first trained a logistic regression model on just the "PloS" data[8]. In training this model we used the elastic net

regularization procedure with alpha=0.99 to produce strong regularization but prevent strongly co-linear genes being excluded. This model was fit using the "glmnet" R package[48].

To obtain regression coefficients specific to each cluster in our training data we fit a series of N binomial logistic regression models, where N is the number of clusters in the training data (i.e., one-versus-rest binomial logistic regression). To prevent the observed frequencies of cells (which we do not expect to accurately reflect the true abundances in situ) from biasing the regression coefficients we use

**Fig. 6 Adult kidney tumors. A** Bulk renal cell carcinomas compared to mature kidney: The relative contribution of single cell derived signals from fetal kidney in explaining the bulk transcriptomes of 171 normal kidney biopsies, 500 clear cell renal cell carcinomas (ccRCC), 274 papillary renal cell carcinomas (pRCC), and 81 chromophobe renal cell carcinomas (ChRCC), along with control populations. The relative contribution of each signal to each bulk RNA-seq sample is shown by the y-axis. Each signal/sample combination is represented by a single point and boxplots shows the distribution with median (middle line), 1st and 3rd quartiles (box limits) and 1.5 times the inter-quartile range (whiskers). Each signal type is abbreviated and colored as per the legend, with squares for fetal and circles for mature. **B** Mast cell fraction in single cell RCC samples: Bar height indicates mast cell fraction (black) or other cell fraction (gray) in 5 single cell RCC expriments (x-axis labels). **C** Mast cell signals in bulk RCC transcriptomes: Inverse of mast cell fraction for bulk transcriptomes (dots) of type given on x-axis. Boxplots show the distribution median (middle line), 1st and 3rd quartiles (box limits), and 1.5 times the inter-quartile range beyond the box-limits (whiskers) and the star indicates that mast cell signals are significantly higher in pRCC T1 type tumors than pRCC T2 (two-sided Wilcoxon rank-sum test, $p = 1.5 \times 10^{-5}$). **D** smFISH validation: An example section of single molecule fluorescence in-situ hybridization imaging of a pRCC T1 tumor section. Nuclei are stained blue with dapi and expression of the tumor marker *MET* is shown in green and the mast cell marker *TPSB2* in purple. See Supplementary Table 4 for a quantification of smFISH applied to pRCC T1/T2 and ccRCC tumors. smFISH imaging was performed on one tumor section from each of pRCC T1, pRCC T2, and ccRCC. The scale bar (bottom-right) indicates approximately 100 μm. **E** Bulk chromophobe renal cell carcinomas compared to mature kidney: The same as **A**, but for 81 chromophobe renal cell carcinomas (ChRCC) bulk transcriptomes. Source data are available as a Source Data file.

---

an offset for each model given by,

$$\log\left(\frac{f}{1-f}\right)$$

where f is the fraction of cells in the cluster being trained.

In each case, we performed 10-fold cross validation and selected the regularization co-efficient, lambda, to be as large as possible (i.e., as few non-zero coefficients as possible) such that the cross validated accuracy was within 1 standard deviation of the minimum.

These models were then used to calculate a predicted similarity for each cell in the combined fetal kidney data set. In calculating the predicted values, an offset of 0 was used. Softmax normalization was not used to allow for the possibility that cell types were present in the combined reference not present in the "PLoS" map. Clusters with a similarity of less than 1 (logit scale) to any of the reference data were labeled as "undecided".

Following the application of the logistic regression model, we elected to merge categories in the reference data that were commonly found in the same clusters. Specifically, we combined:

NPCa, NPCb, and NPCc categories into CapMes.
RVCSBa, RVCSBb into RVCSB.
ErPrT and SSBpr into ErPrT

We removed Leu, Prolif, PTA, and Mes as no cluster contained a majority of these cells.

Each cluster was then annotated with whichever of the reference categories had the highest similarity score averaged across all cells in the cluster. This procedure left one cluster as "Undecided" (that is, most cells in these clusters could not be allocated unambiguously to one of the reference populations). Closer inspection of this cluster revealed in to be an early mesenchymal population, which we labeled as MPC for mesenchymal progenitor cells as discussed in the manuscript.

*Additional reference signal sets.* In addition to the above annotated single cell data sets, cellular reference signals were also taken from additional data sets:

1. A mature kidney single cell reference map[1].
2. The 10x demonstration PBMC data set[49], annotated as described here (https://satijalab.org/seurat/v3.0/pbmc3k_tutorial.html). This data set was used to define a set of leucocyte signals that were added to all other reference maps.
3. Fetal adrenal reference map[12]
4. Whole embryo mouse data[13]
5. A pan-tissue human reference from the human cell landscape publication[24]

*Annotation of congenital mesoblastic nephroma single cell data.* Single cell transcriptomes derived from a congenital mesoblastic nephroma were processed into clusters as described above. Following clustering, we assigned a cell type to each cluster using marker genes identified as in the previous work[1,43].

To confirm that the cluster without expression of other known markers represented CMN tumor cells, we investigated the expression of ETV6, NTRK3, and EGFR in this cluster. CMN is known to be driven by an activating rearrangement between ETV6 and NTRK3, which results in a fusion product with the 5′ end of ETV6 fused to the 3′ end of NTRK3. As the 10X assay measures expression using a 3′ enrichment strategy, CMN tumor cells should show a high level of NTRK3 expression. This is indeed what we find, with the cluster marked tumor expressing NTRK3 more than 10 times more strongly than the next closest cluster. By contrast, ETV6 was not highly expressed in this cluster, as would be expected. Finally, EGFR is known to be highly expressed by tumor cells and was also found to be highly expressed in the cluster designated as tumor.

*Annotation of congenital malignant rhabdoid tumor cell data.* For MRT single cell data, clusters were determined and marker genes identified as described above for CMN. Common non-tumor populations were annotated based on well-known markers (Supplementary Fig. 10). Tumor cells were identified by loss of the SMARCB1, except for MRT2 for which the molecular diagnostic workup did not identify mutation of SMARCB1. For this sample, tumor cells were identified based on the loss expression of other members of the SWI/SNF SMARCA2 and SMARCA4.

*Cell similarity inference using single cell data.* To measure the similarity of a target single cell transcriptome to a reference single cell data-set we used the methodology based on logistic regression outlined in detail[1]. Briefly, we train a logistic regression model with elastic net regularization (alpha = 0.99) on the reference training set. We then use this trained model to infer a similarity score for each cell in the query data set for each cell type in the reference data.

Softmax normalization was not used to allow for the possibility that some cells in the query data set do not resemble any of the cell types in the reference data set. Predicted logits were averaged within each cluster in the query dataset. This approach was implemented using the "glmnet" package in R[48].

*Similarity to mouse data.* In assessing the closest match to the organoid MRT data, we considered a comparison to a mouse reference dataset[13]. We performed the similarity analysis as described above, with the only difference being that we limit the analysis to orthologous genes as determined by ENSEMBL biomart.

*Sensitivity and specificity of samples to particular cellular signals.* To perform the sensitivity and specificity analysis (Fig. 7A, B) we first constructed a set of cellular signals that were indicative of a particular tumor type. These were MPCs and CMN, Intercalated cells and ChRCC, nephrogenic cells (i.e., cap mesenchyme, primitive vesicle, and ureteric bud) and Wilms tumor, PT1 and ccRCC/pRCC, and mature vasculature and ccRCC. We calculated all of these scores for every sample in our data set and then evaluated the sensitivity and specificity at different cut-offs for each score to construct the sensitivity/specificity curves in Fig. 7A (ROC curve).

*Quantification of smFISH images.* The tiled images exported from the Phenix were illumination corrected and stitched together into large, multi-channel fluorescence images by a specialized tool supplied by Perkin-Elmer. These images were then analyzed in the Qupath[50] Bioimage Analysis program. First, all nuclei were segmented using Qupath's cell detection algorithm, then each nucleus was expanded by 3 microns to estimate the area covered by the entire cell. The subcellular spot detection option was then used to detect all fluorescent spots in all the RNA-SCOPE channels for the size range of 1–8 square microns. Each spot was automatically assigned to a detected cell. The detection data for each image was then exported as a.csv file for further analysis.

### Method to quantify single cell-derived signals in bulk transcriptomes

*Data preparation: Bulk RNA-seq data.* Each bulk RNA-seq sample required two pieces of information: fragment counts per gene and the effective length of each gene. As described above, gene counts for this paper were mostly generated using the Rail-RNA pipeline[39] and effective gene lengths as the sum of unique exonic bases per gene. However, fragment counts and effective gene lengths can be calculated in any way.

*Data preparation: Single cell reference data.* To calculate reference single cell signals, single cell data must first be clustered and annotated. Cells are then grouped together by annotation and raw counts summed across all cells within a group. Summed counts are then normalized to sum to 1 across all genes so that a reference

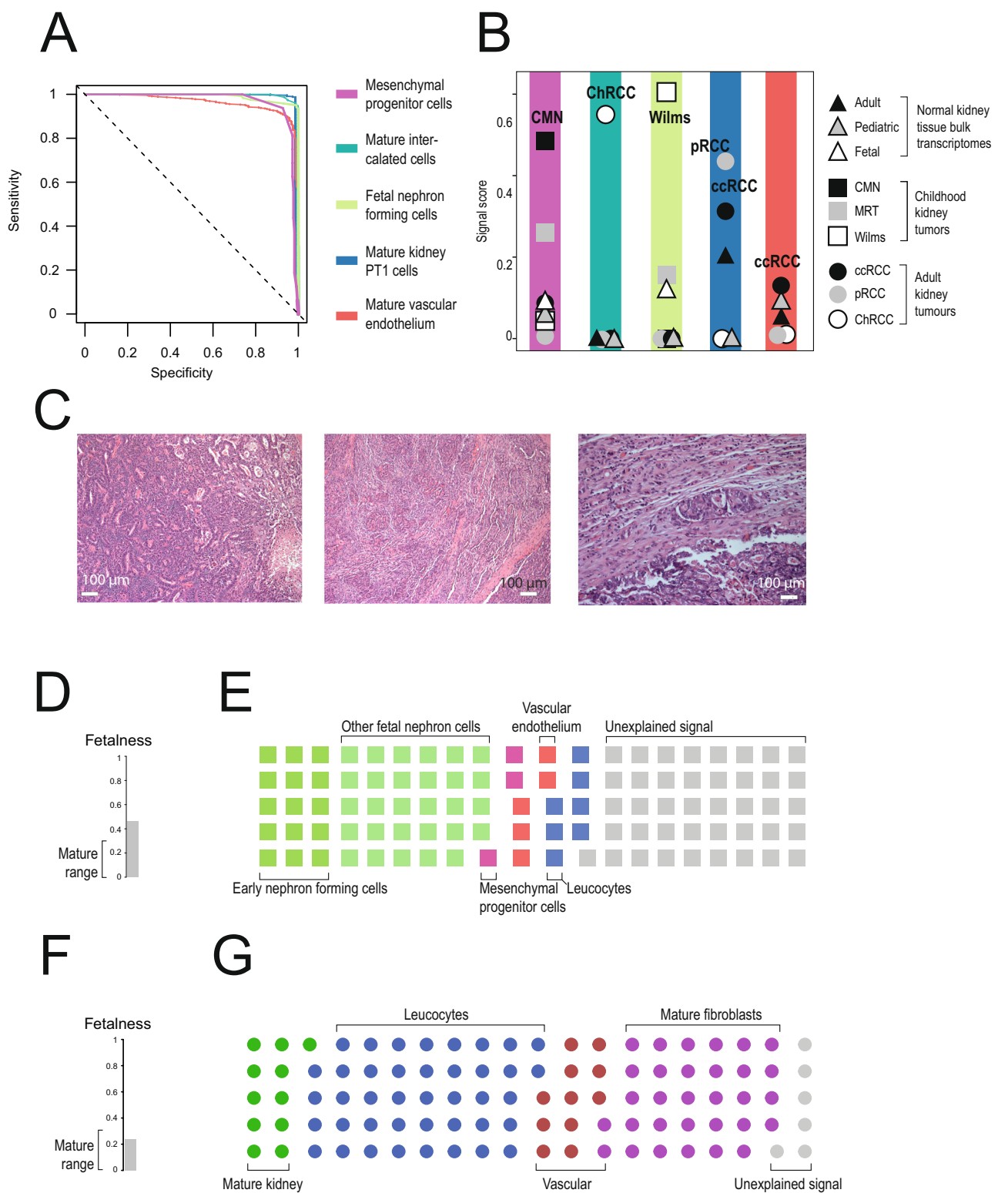

signal is defined as a vector, S,

$$S = \{s_1, s_2, \ldots, s_m\} \, s.t. \sum_{g=1}^{m} s_g = 1$$

where there are m genes.

Reference signals can be constructed in any way (e.g., including batch correction for combined data sets), so long as the final signal can be normalized such that $\sum_g s_g = 1$.

*Model fit.* The aim of the signal assignment method is to infer how much of each of the different reference signals best explains the supplied bulk transcriptome. That is, we are aiming to solve for the values of beta in,

$$y_{gp} = \beta_{0p} + \beta_{1p} s_{g1} + \beta_{2p} s_{g2} + \cdots + \beta_{np} s_{gn}$$

where $y_{gp}$ is the fragment counts for gene g in sample p, $s_{gc}$ is the reference signal c for gene g and $\beta_{cp}$ is the contribution of signal c to sample p. Note that the term $\beta_{0p}$

**Fig. 7 Clinical utility of cellular signal analysis. A** Sensitivity/Specificity of signals in classifying tumor types: Curves showing the sensitivity and specificity of using the scores defined by the color scheme to classify tumors by type at different cut-offs. The different score and tumor type pairs are: fetal interstitial cells and CMN (light blue), intercalated cells and ChRCCs (dark blue), developing nephron and Nephroblastoma (light green), PT1 and ccRCCs/pRCCs (dark green), and mature vascular and ccRCCs (red). **B** Median reference contribution by tumor type: Each point represents the median score for the group of samples indicated by the combination of shape (tissue type, see legend) and shading (score type, as in **A**). **C** Histology image of unclassified childhood renal tumor: The tumor mostly compromised pleomorphic epithelioid cells that formed tubules, papillae, glands and nests, as well as more solid areas with spindled cells and clefting similar to that of synovial sarcoma. Patchy tumor necrosis was apparent. Some areas showed smaller, more uniform cells lining narrow tubular structures, resembling adenomatous perilobar nephrogenic rests. Overall, the morphology and ancillary tests were inconclusive. Scale bars at bottom of each image indicate approximately 100 μm. **D** Immaturity score for unclassified childhood renal tumor: Calculated as in Fig. 2, with score range for normal post-natal kidney indicated on left. **E** Summary of signal contribution from fetal and mature kidney to unclassified childhood renal tumor: Each color represents the signal type labeled and fraction of squares of each type matches the signal contribution. **F** Immaturity score for childhood renal cell carcinoma: As in **D**. **G** Summary of signal contribution from fetal and mature kidney to childhood renal cell carcinoma: As in **E** but for a transcriptome derived from renal cell carcinoma fit using a mature kidney signal set. Source data are available as a Source Data file.

represents the contribution the intercept term for sample p, which is equivalent to the inclusion of an additional flat signature for which $s_g = \text{constant} \forall g$. The inclusion of this intercept term provides a measure of the extent to which the reference signal set is inappropriate for the sample given.

In order to efficiently calculate the contributions of the same set of reference signals simultaneously, we formulated the following model,

$$\beta_{cp} = e^{z_{cp}}$$

where c represents the signal and p the sample as above. We solve for $z_{cp}$ instead of $\beta_{cp}$ directly as this formulation ensures that the contributions of each signal are always strictly positive. Next, we calculate the expression for gene g and sample p implied by these values of $\beta_{cp}$

$$\lambda_{gp} = S_{gc}\beta_{cp}$$

where g represents the gene, and then calculate

$$\lambda'_{gp} = \lambda_{gp} l_{gp}$$

where $l_{gp}$ represents the effective length of gene g in sample p. This modification by length is necessary as the fragment counts by gene created by bulk RNA-seq are proportional to the length of a gene. Finally, a joint negative log likelihood is calculated as

$$-\log \mathscr{L} = \sum_p \sum_g w_g (\lambda'_{gp} - y_{gp} \log \lambda'_{gp})$$

where $w_g$ is an optional gene penalty applied to genes deemed to be biologically less important. In this paper we set $w_g$ to 1 for all genes except a set of housekeeping genes that are set to 0.5 and metabolic genes set to 0. It is this equation that is minimized with respect to $z_{cp}$ in order to solve for the values of $\beta_{cp}$. This log-likelihood is simply the Poisson log-likelihood for a Poisson distribution with mean $\lambda'_{gp}$ (see section below for a discussion of this choice of distribution).

This model is directly specified using the tensorflow framework[51], which allows the efficient minimization of this equation utilizing graphics processors and multi-core machines. The optimization is performed using stochastic gradient decent using Adam[52]. We require two termination conditions to be met before the optimization is terminated:

1. the fractional decrease in the log-likelihood must be less than some tolerance parameter.
2. the fractional change in Q must be less than the same tolerance parameter.

Q is defined as the sum of the sigmoid transformation of $z_{cp}$ and roughly measures the number of signals with non-zero contributions to the fit. Without this second termination condition, optimization would terminate with the coefficients of many signals given small but non-zero values as these non-zero values barely shift the total log-likelihood.

*Post processing.* Having obtained optimized values for $\beta_{cp}$, we next normalize these values by first modifying the intercept term to

$$\beta'_{0p} = \frac{\beta_{0p}}{m}$$

which make the modified intercept term equivalent to fitting an additional signal with a completely flat profile. Following this modification, we then normalize the values of $\beta_{cp}$ to sum to 1 for each sample. These normalized values represent the relative contribution of each signal to each sample and are the values reported throughout this manuscript. The final normalization step essentially controls for differences in bulk RNA-seq library size and makes the coefficients comparable across samples.

As an additional measure of the goodness we re-fit the above model with only the intercept term and then calculate,

$$pR^2 = 1 - \frac{\log \mathscr{L}_{full}}{\log \mathscr{L}_{int}}$$

which is, a McFadden's pseudo R-squared value[53] $pR^2$ given by 1 minus the ratio of the log-likelihood of the full model fit over the model fit with only the intercept term.

In order to aid with interpretation of the fit, the contribution from similar cellular signals is often aggregated before being presented in the Figures and Supplementary Figures. For example, there are multiple endothelial signals in the mature kidney, but for simplicity and readability we have combined them throughout this study.

*Quantification of goodness of fit.* One of the central aims of cell signal analysis compared to deconvolution methods is to allow for the possibility of a mismatch between the reference and the cells present in the bulk transcriptome. We allow for this through the inclusion of an intercept term in the model and quantify the mismatch by the relative contribution of the intercept and the fraction of the variance explained by the reference as measured by $pR^2$. To understand the intuition behind this choice it is useful to consider a simple linear model with and without an intercept.

Supposing we have bulk transcriptomes composed of two cell populations A and B. As a reference we have single cell transcriptomes from population A, but not B. Let us now consider how a linear model with and without an intercept term behaves. That is, for each bulk transcriptome we fit two models:

$$Y_g = \beta_A R_g \tag{1}$$

$$Y_g = \beta_A R_g + \beta_0 \tag{2}$$

Where $Y_g$ is the expression in the bulk transcriptome for gene g, R is the reference signal at gene g derived from single cell transcriptomes population A, and the ß terms are estimated using linear regression. If $Y_g$ is composed only of cells from population A, models (1) and (2) are identical (i.e., $\beta_0 = 0$). Let us now consider what happens if a small number of cells from population B are added to $Y_g$. Further, assume that cell types A and B are unrelated cell types with uncorrelated transcriptomes. The maximum likelihood estimator for beta in model (1) is:

$$\hat{\beta}_A = \frac{<YR>}{<R^2>}$$

where <…> denotes an average across all genes. For model (2) the maximum likelihood values of ß are:

$$\hat{\beta}_A = \frac{<YR> - <Y><R>}{<R^2> - <R>^2} = \frac{cov(Y,R)}{cov(R,R)}$$

$$\hat{\beta}_0 = <Y> - \hat{\beta}_A <R>$$

where cov(…) indicates the covariance of the two variables. Write $Y_g = n_A R_g + n_B S_g$, where $S_g$ is the cell signal reference for population B and $n_A$ and $n_B$ give the number of cells from each population. It can then be shown that the difference in ß in the simple population A only model ($n_B = 0$) and the full model with both populations is the same as fitting a model with $n_A = 0$. That is,

$$\Delta \hat{\beta}_A = n_B \left( \frac{<SR> - <S><R>}{<R^2> - <R>^2} \right) = \frac{cov(S,R)}{cov(R,R)}$$

$$\Delta \hat{\beta}_0 = n_B \left( <S> - \Delta \hat{\beta}_A <R> \right)$$

Where delta represent the difference between the model with and without a contribution from population B (i.e., the difference between models where $n_B = 0$ and $n_B > 0$). The equivalent formula for model (1) is,

$$\Delta\hat{\beta}_A = n_B \left( \frac{<SR>}{<R^2>} \right)$$

From this it can immediately be seen that the effect of adding a cell population not accounted for in the reference (population B in this example) in model (1) is to increase the contribution from the unrelated population present in the reference (population A, $\Delta\beta_A > 0$). By contrast, for model (2) the contribution from the unrelated reference population (A) remains unchanged ($\Delta\beta_A = 0$) and the intercept term is increased to model the unaccounted-for population (B).

Of course, our model is not a linear model, but is based on a constrained generalized linear model. Furthermore, the above argument assumes that population B is unrelated to A. If population B has some correlation/covariance with A, then the fit for the population with which it is correlated will be increased by an amount proportional to the degree of correlation (see formula above). However, it is straightforward to show that (assuming Y and R are strictly positive) model (2) will always increase $\beta_A$ by less than model (1). Thus, the inclusion of the intercept term will always improve the ability of the model to handle mismatches between the reference cell signals and the transcriptomes of the cells that make up each bulk transcriptome.

One thing the above toy example makes clear is that while the intercept term accounts for and quantifies variation not in the reference to an extent, it is not a complete solution. To further identify cases where there is a mismatch between the reference and the observed data we make use of the goodness of fit statistic, $pR^2$ define above. Continuing with the linear model analogy, this metric will identify cases where Y is modified by a perturbation δ that can both increase and decrease expression. An example of this would be the changes to a normal cell's transcriptome as it transforms into cancer. Suppose that the perturbation was not correlated with the reference and has an average value of zero. It follows that the effect of this perturbation on the model fit parameters is,

$$\Delta\hat{\beta}_A = \frac{cov(\delta, R)}{cov(R,R)} = \frac{0}{cov(R,R)} = 0$$

$$\Delta\hat{\beta}_0 = n_B \left( <\delta> - \Delta\hat{\beta}_A <R> \right) = n_B(0 - 0 <R>) = 0$$

However, although the fit does not change, the perturbation will decrease the total likelihood of the model and increase the value of $pR^2$. In combination, the intercept term and pseudo R-squared metric provide a quantification of the mismatch between reference cell signals and bulk transcriptomes

*Benchmarking.* We compared our method to two other methods: MuSiC[15] and BSeq-SC[14]. In both cases, we used the default settings recommended by each method. As MuSiC requires a reference containing multiple cells derived from multiple samples, we were unable to create include leucocytes as part of our reference panel for Fig. 1D, E for MuSiC. BSeq-SC required marker genes for each population in the reference. To generate markers for each reference population we identified genes significantly enriched in the target population with a binomial test, using the quickMarkers function in the SoupX R package[43]. We further refined this set of markers by requiring that markers be expressed in at least 40% of cells within the cluster they mark and less than 5% of all other clusters. With these requirements, some clusters, notably the PT1 cluster, did not have any marker genes and so received a value of 0 in the BSeq-SC fit.

*Calibration of intercept term.* To assess the range of contributions from the intercept term (i.e., the "unexplained signal") when no appropriate reference is provided we constructed a series of inappropriate fits. In each case, we selected a cell signal reference set that we knew was inappropriate to the set of samples being considered. The range of intercept values in this fit then gives a quantitative range that is indicative of how much weight is given to the intercept when no appropriate reference is present.

*Choice of Poisson distribution.* The choice of the Poisson distribution as the likelihood model at first glance seem a curious one, given that the Negative Binomial distribution (of which the Poisson distribution is a specific case) is widely used to model both bulk RNA-seq and single cell RNA-seq data. However, some reflection reveals that this choice is actually well justified.

We wish to evaluate the probability of observing a particular number of fragment counts in a bulk RNA-seq experiment, given that this experiment is composed of the addition of signals from a collection of single cell derived transcriptomic signals. To do this, we need to know how likely a particular set of fragment counts is, given a fixed contribution from each of the single cell signals.

Let us assume that the number of counts for a gene g in cell type c in a single cell RNA-seq experiment (with a fixed number of reads) can be well modeled by a negative binomial distribution with mean μ and over-dispersion φ. That is, the

variance of this distribution is given by,

$$\sigma_{gc}^2 = \mu_{gc} + \mu_{gc}^2 \phi_{gc}$$

The distribution we are interested in, is then the distribution resulting from the sum of $\{N_0, N_1, N_2, \ldots, N_k\}$ random samples from the set of negative binomial distributions representing cell types $\{1,2,\ldots,k\}$. That is, the distribution we are interested in is given by the sum of negative binomial distributions.

The moment generating function for the compound distribution is then given by,

$$M_{comp}(t) = \prod_{c \in C} \left( 1 + \mu_c \phi_c \left( 1 - e^t \right) \right)^{-\frac{1}{\phi_c}}$$

where C is the set of signals summed to form the compound distribution. This moment generating function completely specifies the compound distribution. However, we can use the method of moments to approximate this compound distribution with another Negative Binomial distribution with mean μ and over-dispersion φ. To do this, observe that the first and second moment of the compound distribution are,

$$E(X) = \sum_{c \in C} \mu_c$$

$$E(X^2) = \sum_{c \in C} \mu_c + \left( \sum_{c \in C} \mu_c \right)^2 + \sum_{c \in C} \mu_c^2 \phi_c$$

while the first and second moments of a negative binomial distribution with mean μ and over-dispersion φ are,

$$E(X) = \mu$$

$$E(X^2) = \mu + \mu^2 + \mu^2 \phi$$

Using the method of moments, this implies that the negative binomial approximation to the compound distribution has,

$$\mu = \sum_{c \in C} \mu_c$$

$$\phi = \sum_{c \in C} \left( \frac{\mu_c}{\mu} \right)^2 \phi_c$$

That is, the mean of the compound distribution equals the sum of the means of each component distribution (as expected). The over-dispersion of the compound distribution is equal to the weighted sum of the component distributions. Closer consideration of the equation for the compound over-dispersion reveals that the over-dispersion of the compound distribution is almost always considerably less than the average over-dispersion of its component distributions.

For example, consider the case where all distributions have approximately the same mean and rewrite the compound over-dispersion as,

$$\phi = \sum_{c \in C} \left( \frac{\mu_c}{<\mu_c>} \right)^2 \frac{\phi_c}{N^2}$$

where angle brackets denote an average and N is the number of elements in C. Assuming the ratio in brackets is close to 1 gives,

$$\phi = \frac{<\phi_c>}{N}$$

So, in the case of distributions with similar means, the over-dispersion of the compound distribution is always N times less than the mean over-dispersion of the individual distributions. Consequently, as the number of distributions being summed over increases, the over-dispersion goes to zero and the compound Negative Binomial distribution approaches a Poisson distribution. The more general case where the means of the component distributions are not all similar is more complex, but in the limit of many distributions, the compound over-dispersion still approaches 0.

This result justifies the use of a Poisson distribution as the likelihood model in our fitting procedure. Although the individual signals from which the fit is derived are negative binomially distributed, the distribution of their sum is Poisson distributed. It may be that an extension of the Poisson model used here may prove useful, to model effects such as uncertainty in the effective length of genes for example, but it is not required to accurately represent the compound distribution on which our model depends.

**Reporting summary**. Further information on research design is available in the Nature Research Reporting Summary linked to this article.

## Data availability

Raw nucleotide sequences for single cell data newly generated for this study are available in the European Genome-Phenome Archive under restricted access with accession codes EGAD00001004304 (CMN), EGAD00001006296 (MRT organoids), EGAD00001007498 (CCSK CEL-Seq2), and EGAD00001007572 (Wilms, other CCSK, and other MRT), access can be obtained by contacting the Data Access Committees EGAC00001001146 (CCSK CEL-Seq2) or EGAC00001000205 (everything else). We additionally utilised

publicly available single cell kidney data from previous work[1,8,54]. The raw nucleotide data for which are available in the European Genome-Phenome Archive under restricted access with accession codes EGAS00001002171, EGAS00001002486, EGAS00001002325, and EGAS00001002553, and in the Human Cell Atlas Data Portal with project ID abe1a013-af7a-45ed-8c26-f3793c24a1f4.

Raw nucleotide sequences for bulk transcriptomic data newly generated for this study are available in the European Genome-Phenome Archive under restricted access with accession codes EGAS00001002487 and EGAS00001002534, access can be obtained by contacting the Data Access Committee EGAC00001000205. We also utilised publicly available bulk transcriptomes from: congenital mesoblastic nephroma[22], Wilms tumor[55], fetal kidney[56], the Therapeutically Applicable Research to Generate Effective Treatments (https://ocg.cancer.gov/programs/target) initiative, phs000218, available at https://portal. gdc.cancer.gov/projects, data generated by the TCGA Research Network, available at https://www.cancer.gov/tcga, and The Genotype-Tissue Expression (GTEx) Project as mapped by the recount2 project[38].

Mapped count data (i.e., tables of counts) are available as Supplementary Data 2 (bulk transcriptomes) and Supplementary Data 3 (single cell transcriptomes). This includes both newly generated data in this study and data obtained from public repositories. Sample metadata, including references to the source from which this data was obtained are listed for each unit of data in Supplementary Data 1 (bulk transcriptomes) and Supplementary Table 2 (single cell transcriptomes). All patient samples generated in this study are listed in Supplementary Table 5. The remaining data are available within the Article, Supplementary Information or Source Data file. Source data are provided with this paper.

## Code availability

As an annex to the supplementary methods, we have provided all source code used in generating the results, figures, and tables used in this study as Supplementary Software 1. The purpose of these code files is to provide additional details as to how we implemented the analyses described in the Methods section. We provide the code necessary to run cellular signal analysis, along with some documentation and an example dataset online at https://github.com/constantAmateur/cellSignalAnalysis.

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

## Acknowledgements

We acknowledge funding from Wellcome (Fellowships to S.B., K.S., J.C.A. 209328/Z/17/Z, core funding to the Sanger Institute, strategic award 211276/Z/18/Z). Additional support was provided by the St Baldrick's Foundation (S.B.), Great Ormond Street Hospital Children's Charity (J.C.A.), Great Ormond Street Hospital Biomedical Research Centre, Olivia Hodson Cancer Fund (K.S.), CRUK (Fellowship to T.J.M.), National Institute for Health Research funded Cambridge Biomedical Research Centre (R.A.B., A.Y.W). The views expressed are those of the authors and not necessarily those of the National Health Service, National Institute for Health Research, or Department of Health.

The CUH adult renal cancer sampling had infrastructure support from the Urological Malignancies Programme which is part of the CRUK Cambridge Centre, funded by Cancer Research UK Major Centre Award C9685/A25117, and supported by the NIHR Cambridge BRC. The views expressed are those of the authors and not necessarily those of the NIHR or the Department of Health and Social Care. F.C.H. acknowledges funding from the ERC advanced grant DynaMech-671174. We thank the Máxima FACS facility for sorting, Single Cell Discoveries for library preparations, and Single Cell Genomics facilities for help with scRNA analysis. J.D. acknowledges funding from the European Research Council (ERC) starting grant 850571, Dutch Cancer Society (KWF/Alpe d'HuZes Bas Mulder Award; KWF/Alpe d'HuZes, #10218), Foundation Children Cancer Free (KiKa #338, L.C.), Oncode Institute. We thank Dr Amos Burke for his contribution. We are grateful to our patients, young and old, for participating in our study.

## Author contributions

M.D.Y. and S.B. conceived of the experiment and wrote the manuscript. M.D.Y. performed analyses, aided by T.J.M., E.K., G.K., and T.H.H.C. I.D.V. and J.C.A. provided expertise on adrenal gland analysis. L.C. performed organoid experiments with F.A.V.B. T.R.W.O., N.S., D.R., N.C., L.H., R.R.K., A.W. provided pathological expertise. F.C., M.M.H.E., and A.S. provided clinical data. A.P., E.B.B., F.M., C.T., C.B., G.D.S., V.J.G., M.H., M.K., S.M.P., O.A.B., K.R., K.K, F.C.H., J.D., F.C.H., E.P., K.A., contributed to discussions and/or data. S.A.T., T.M., F.C.H., F.M., J.D., R.R.K, contributed fetal and tumor single cell data, together with K.B.M., R.A.B., X.H., A.W.C, L.M. S.B., and M.D.Y. directed the study, in conjunction with K.S. (single cell cancer work) and J.D. (organoid work).

## Competing interests

The authors declare no competing interests.

## Additional information

[1]Wellcome Sanger Institute, Wellcome Genome Campus, Hinxton, Cambridge, UK. [2]Cambridge University Hospitals NHS Foundation Trust, Cambridge, UK. [3]Department of Surgery, University of Cambridge, Cambridge, UK. [4]Princess Máxima Center for Pediatric Oncology, Utrecht, The Netherlands. [5]Oncode Institute, Utrecht, The Netherlands. [6]Department of Pathology, University Medical Center Utrecht, Utrecht, The Netherlands. [7]Program in Genetics and Genome Biology, The Hospital for Sick Children, Toronto, ON, Canada. [8]UCL Great Ormond Street Hospital Institute of Child Health, London, UK. [9]Amsterdam UMC, University of Amsterdam, Amsterdam, The Netherlands. [10]Cambridge Urology Translational Research and Clinical Trials office, Cambridge Biomedical Campus Cambridge CB2 0QQ University of Cambridge, Cambridge, UK. [11]Great Ormond Street Hospital for Children NHS Foundation Trust, London, UK. [12]NIHR Great Ormond Street Hospital BRC and Institute of Child Health, London, UK. [13]Department of Pathology, University of Cambridge, Cambridge, UK. [14]Department of Dermatology and NIHR Newcastle Biomedical Research Centre, Newcastle Hospitals NHS Foundation Trust, Newcastle upon Tyne, UK. [15]Intitute of Cellular Medicine, Newcastle University, Newcastle upon Tyne, UK. [16]Hopp Children´s Cancer Center Heidelberg (KiTZ), Heidelberg, Germany. [17]German Cancer Research Center (DKFZ) and German Cancer Consortium (DKTK), Division of Pediatric Neurooncology, Heidelberg, Germany. [18]Heidelberg University Hospital, Department of Pediatric Hematology and Oncology, Heidelberg, Germany. [19]MRC-WT Cambridge Stem Cell Institute, University of Cambridge, Cambridge, UK. [20]Department of Clinical Neuroscience, University of Cambridge, Cambridge, UK. [21]Department of Laboratory Medicine and Pathobiology, University of Toronto, Toronto, ON, Canada. [22]Department of Paediatric Laboratory Medicine, The Hospital for Sick Children, Toronto, ON, Canada. [23]Cavendish Laboratory, University of Cambridge, Cambridge, UK. [24]Department of Paediatrics, University of Cambridge, Cambridge, UK. [25]These authors contributed equally: Matthew D. Young, Thomas J. Mitchell, Lars Custers. [26]These authors jointly supervised this work: Matthew D. Young, Jarno Drost, Karin Straathof, Sam Behjati. ✉email: my4@sanger.ac.uk; j.drost@prinsesmaximacentrum.nl; k.straathof@ucl.ac.uk; sb31@sanger.ac.uk

