## [Peer Review File · Nature Communications]

Reviewer #4, expert in single cell sequencing and deconvolution (Remarks to the Author):

I will limit my comments to the bioinformatics analysis. I am glad to say that it seems mostly sensible. There are a few parts that are bemusing - for example, the use of the MCVR method to choose the optimal number of PCs seems like an unnecessary complication - but I don't think those parts matter all that much.

The deconvolution strategy looks reasonable enough. I have some concerns about their use of the intercept as a quality control measure for the appropriateness of the fit. It seems very difficult for this value to increase in response to, e.g., the presence of a cell type that is missing from the reference. To do so requires a consistent additive effect across all genes in the sample, and it is easy to imagine situations where this does not occur. For example, if almost all marker genes are fitted perfectly with the intercept set to zero, there is no scope for the model to increase the intercept in response to a handle of markers for a missing cell type. Similarly, it is not clear to me that a non-zero intercept in the model is sufficient to restore accuracy for the remaining cellular signal estimates in the presence of a cell type/signal that is not in the reference.

I am also surprised that existing deconvolution methods do not work, such that the authors were compelled to develop their own bespoke method to analyze this dataset. This usually requires a demonstration that their new method has at least comparable performance to the existing methods on some independent datasets (beyond the RCC dataset presented in the manuscript). I could not find any such results in either the main manuscript or the supplementary materials. Were I being uncharitable, I would say that there is a 50:50 chance that any novelty in the authors' results are an artifact of their method, rather than any failure of the existing methods. Some additional testing would put these concerns to bed.

Finally, while this is outside the scope of my review, I will second reviewer 3's comments that this is quite a descriptive study. It would have been nice to see a more extensive follow-up of just one of the computational results in this manuscript. As it stands, it's like I'm reading four beginnings of a story rather than a cohesive narrative, which leaves the reader with just as many questions as answers. For example: do we see this effect in a wider population of CCSK patients beyond a single sample? Can the putative mesenchymal origin be validated with lineage tracing? Same for the MRTs? Does this approach have diagnostic benefit beyond a single 11yo boy? And so on.

Reviewer #5, expert in kidney cancer biology and genomics (Remarks to the Author):

In this paper, the author analysed transcriptomic data from a large number of renal tumor samples with a focus on characterising the different cellular origin of childhood and adult tumors. They deconvoluted the contribution of different cell types toward the measured transcriptomic signal using single cell transcriptomic from the Human Cell Atlas with a new method. They show that this method performs better for this task than other deconvolution methods and they also conclude that childhood tumors are of fetal origin whereas adult tumors are of normal kidney cell origin.

Overall, the study is highly interesting and the usefulness of the deconvolution method is clearly demonstrated for the data presented. The conclusion regarding the cell origin of tumors and dedifferentiation state are of high interest for the field. This method could benefit cancer studies in different contexts. Authors have responded well to all concerns raised by the previous reviewers. There are some minor points (detailed below) that would need to be clarified before publication.

In the manuscript, the authors conclude that the tumor cells of adult kidney tumors are not globally dedifferentiated. This conclusion is very important but is currently not described well in the main text. The term "dedifferentiation" can be interpreted in different ways. The concept is actually much better explained in the rebuttal, where the authors clearly describe the concept of dedifferentiation as "reversion of adult tumours to a foetal state at the whole transcriptome level". It would be great if the authors could also mention this clearly in the main text of their manuscript.

The authors shows interesting examples on how this approach can be used to generate diagnostic clues. While the results are certainly encouraging, only two examples are not enough to tell how

accurate and generalizable the method may be for such use (only 2 patients). We would recommend changing the title of the section "Single cell signal provides diagnostic clues" to reflect this, and explain this further in the discussion.

In the discussion, the authors state that they observe "Remarkably uniform cellular signals". It is not clear to which results exactly the authors refer to when making this statement. It would also help if the authors could give and discuss specific examples of such features from the data.

Are there plans to make this new method to deconvolute signal contribution as a package? In its current state, it is not clear how other scientists may use this method with their own data. Furthermore, what are the requirements and limits of the method? For example, how many cells / samples are required to obtain accurate deconvolution ? I think making this tool available will certainly be important for the field!

It would be useful to add titles to the plots of figure 7 E and G to indicate what they represent.

Rafael Kramann

Revision of NCOMMS-20-44567A = Single cell derived mRNA signals across human kidney tumors

Reviewer #4, expert in single cell sequencing and deconvolution (Remarks to the Author):

4.0	I will limit my comments to the bioinformatics analysis. I am glad to say that it seems mostly sensible	Thank you.
4.1	There are a few parts that are bemusing - for example, the use of the MCVR method to choose the optimal number of PCs seems like an unnecessary complication - but I don't think those parts matter all that much.	We do not entirely understand the objection here. The standard practice in the field is to arbitrarily choose a number of PCs based on (at best) “squinting” at a plot of variance explained by number of PCs. The molecular cross validation method proposes a mathematically justified approach to avoid this arbitrary choice and motivate the selection of number of PCs (https://www.biorxiv.org/content/10.1101/786269v1). We agree that this is a relatively minor point and will not change the findings of the paper.
4.2	The deconvolution strategy looks reasonable enough. I have some concerns about their use of the intercept as a quality control measure for the appropriateness of the fit. It seems very difficult for this value to increase in response to, e.g., the presence of a cell type that is missing from the reference. To do so requires a consistent additive effect across all genes in the sample, and it is easy to imagine situations where this does not occur. For example, if almost all marker genes are fitted perfectly with the intercept set to zero, there is no scope for the model to increase the intercept in response to a handle of markers for a missing cell type. Similarly, it is not clear to me that a non-zero intercept in the model is sufficient to restore	Motivated by the reviewers comments we have: - Included a theoretical exploration of how a model with and without an intercept term will quantify a model where there is no relationship between the covariates and response variable. We can show that in the limit of no relationship between the covariates and the response variable, the fit will only assign non-zero values to the intercept term. In more general circumstances where there is some correlation between the bulk expression and the single cell derived signals, an intercept model will always improve upon a model without an intercept (where improvement is measured as attributing less signal

accuracy for the remaining cellular signal estimates in the presence of a cell type/signal that is not in the reference.

to inappropriate reference populations).

- See the section “Quantification of goodness of fit” in the supplementary methods for a full explanation. As well as the points mentioned above, it also follows that the situation the reviewer describes can be well accounted for with an intercept term. We show that fitting a bulk signal comprised mixture of two populations in which one has a perfect reference, is mathematically equivalent to fitting the second population on its own. Provided this second population is not strongly correlated with the perfectly explained one, the effect of including it will be to increase the value of the intercept.
- To further demonstrate this point, we have included an example of how the intercept behaves as the reference population is made progressively more unsuitable (**Fig. S2C**). We start with a bulk population that is well explained by a single reference cellular signal (B cells). We then progressively perturb the reference population decreasing its suitability as a reference for the bulk data. We show that as the reference becomes more and more unsuitable, the unexplained signal (intercept term) progressively dominates the fit.

Fig. S2C – Bulk transcriptomes of flow sorted B cells are fit using a reference derived from single cell B cells. The y-axis then shows how much signal is attributed to the unexplained signal (Intercept) and how much to the reference. As you move to the right on the x-axis the reference used is randomised to make it less and less appropriate, resulting in a steady increase of the Intercept term.

- We agree that the intercept term alone cannot provide a complete control of the appropriateness of the fit. To complement this metric, we also calculate a pseudo-R squared value, as discussed in the

		supplementary methods in the “post processing” and “Quantification of goodness of fit” sections. In general, there is a strong anti-correlation between the pseudo-R squared value and the unexplained signal. That is, when the unexplained signal is high, the pseudo-R squared value is low. For narrative simplicity, we elected to focus on the unexplained signal as the primary metric in the main manuscript as in our opinion this quantification was most likely to be widely understood. For expert readers such as the reviewer we have included the pseudo-R squared values for all our fits in Fig. S4,6,9,12,14 and Data S3.
4.3	I am also surprised that existing deconvolution methods do not work, such that the authors were compelled to develop their own bespoke method to analyze this dataset. This usually requires a demonstration that their new method has at least comparable performance to the existing methods on some independent datasets (beyond the RCC dataset presented in the manuscript). I could not find any such results in either the main manuscript or the supplementary materials.	We try to be clear that we do not think that existing methods are “wrong”. Indeed, they perform admirably at the task that they are designed for: quantifying the composition of a bulk sample based on a complete set of reference cell types. Our point is that the question we are interested in is one for which they were not designed. We have changed the main text to make this clear (see section “Quantification of reference cellular mRNA signals in bulk transcriptomes”). To further clarify this, we have included a new benchmark evaluating a key metric in “traditional deconvolution”: accurately inferring the correct abundances of cells in a pseudobulk transcriptomes. When evaluated on this metric, our approach (cellular signal analysis) performs worse than the best existing methods, although the accuracy is still good (Fig. S2D).

Fig. S2D – 100 Bulk transcriptomes were generated by randomly summing cells selected from the single cell reference. The accuracy of MuSiC and Cell Signal Analysis in recovering the true cell proportions was then calculated and the root mean squared error is what is plotted above. Solid lines represent the median. This shows that each method performs best on the task for which it is designed. Cell Signal Analysis is best able to account for an inappropriate reference, while deconvolution methods achieve slightly higher accuracy at recovering the correct cellular

composition of bulk tissues when a complete reference is available.

This is the cost to including an intercept term in the statistical model and why existing methods do not do so. Existing methods are interested in solving the problem of estimating the cellular composition where a complete reference is available. Including an intercept in such circumstances will moderately decrease the accuracy of methods designed for this task and so they do not do so.

We are not primarily interested in quantifying how many cells are present in a tumour transcriptome, but instead aim to identify which reference normal population contributes the most to the transcriptome of the cancer cells. For this task, it is more important to have a way of preventing the assignment of inappropriate signals and so including an intercept is a sensible approach.

As well as the pseudobulk benchmark (**Fig. S2D**), we have included benchmarking using bulk transcriptomes from leukaemias of known origin, peripheral blood, and flow sorted leucocytes (**Fig. 1D-E**). In performing these comparisons, we are evaluating not just the ability of each method to recover the correct cell type, but also its success (or otherwise) in handling the discrepancies between the bulk transcriptome and provided single cell derived reference. Further details of these benchmarks are available in **Fig. S2**.

	E Reference signals  Matching cell Other leucocytes Inappropriate signal (Renal PT1) Unexplained Cellular Signal Analysis MuSiC MuSiC must use inappropriate signal when no match in reference Cellular Signal Analysis allocates discrepancy to unexplained signal Unmatched Cells Cancer (ALL) Unmatched Cells Cancer (ALL) Bulk transcriptomes with no exact match in reference Fig 1E – When bulk transcriptomes are fit using a deliberately incomplete reference, MuSiC will assign implausible reference signals (green). This is true whether the mismatch is due to absence of a cell type from the reference (Unmatched cells on left) or because cells are transformed relative to the reference (cancer data on right).
4.4 Were I being uncharitable, I would say that there is a 50:50 chance that any novelty in the authors' results are an artifact of their method, rather than any failure of the existing methods. Some additional testing would put these concerns to bed.	We hope that the additional benchmarking and mathematical arguments provided above are convincing to the reviewer. We would further emphasise that our starting point is always that no one method or piece of data can be completely relied upon. It is for this reason that we have gone to considerable effort to generate validation data (in the form of tumour derived single cell transcriptomes) in our

		manuscript. Several of the tumour types we study are extremely rare and so acquiring the material for such a validation has been extremely challenging. We hope the fact that we have nonetheless taken the effort to generate these data demonstrates that we recognise the importance of validating our findings. We also provide additional layers of validation in several cases: immunohistochemistry, single molecule fluorescence in-situ hybridisation, and examination of patterns of somatic alterations to DNA.
4.5	Finally, while this is outside the scope of my review, I will second reviewer 3's comments that this is quite a descriptive study. It would have been nice to see a more extensive follow-up of just one of the computational results in this manuscript. As it stands, it's like I'm reading four beginnings of a story rather than a cohesive narrative, which leaves the reader with just as many questions as answers. For example: do we see this effect in a wider population of CCSK patients beyond a single sample? Can the putative mesenchymal origin be validated with lineage tracing? Same for the MRTs? Does this approach have diagnostic benefit beyond a single 11yo boy? And so on.	Our investigation provides an overview of cellular signals across the entire spectrum of human renal tumours, akin to a mutational signature analyses of cancer genomes. We completely agree that we have obtained many insights that will have to be pursued in the future, but it really would be beyond the scope of our work to pursue all these insights in a single paper. The focus of our work is the cell signals themselves. The aspect of our paper that we wanted to flesh out was the clinical diagnostic utility, which is convincingly shown in the AUC curves. We agree that it would be interesting to include more ambiguous cases for investigation. Thankfully, such cases are extremely rare, and the overwhelming majority of diagnoses of renal tumours can be reached by an experienced pair of eyes. However, when it cannot, as illustrated by the case of our patient, our method has proved to be diagnostically extremely powerful in terms of quantifying the maturity of the tumour (adult VS foetal) and in terms of phenotyping (i.e. Wilms like).

Reviewer #5, expert in kidney cancer biology and genomics (Remarks to the Author):

5.0	In this paper, the author analysed transcriptomic data from a large number of renal tumor samples with a focus on characterising the different cellular origin of childhood and adult tumors. They deconvoluted the contribution of different cell types toward the measured transcriptomic signal using single cell transcriptomic from the Human Cell Atlas with a new method. They show that this method performs better for this task than other deconvolution methods and they also conclude that childhood tumors are of fetal origin whereas adult tumors are of normal kidney cell origin. Overall, the study is highly interesting and the usefulness of the deconvolution method is clearly demonstrated for the data presented. The conclusion regarding the cell origin of tumors and dedifferentiation state are of high interest for the field. This method could benefit cancer studies in different contexts. Authors have responded well to all concerns raised by the previous reviewers	
5.1	In the manuscript, the authors conclude that the tumor cells of adult kidney tumors are not globally dedifferentiated. This conclusion is very important but is currently not described well in the main text. The term “dedifferentiation” can be interpreted in different ways. The concept is actually much better explained in the rebuttal, where the authors clearly describe the concept of dedifferentiation as “reversion of adult tumours to a foetal state at the whole transcriptome level”. It	We agree, and thank the reviewer for pointing this out. We have made it clear that what we are measuring here is a similarity of the transcriptome not for a handful of key marker genes, but a broad adoption of a developmental pattern. Specifically, we have made the following changes to the manuscript: - Added sentence to section “Childhood tumors, but not adult tumors, exhibit a fetal transcriptome”:

	would be great if the authors could also mention this clearly in the main text of their manuscript.	We define dedifferentiation to be the reversion of a mature cell to a fetal state, at the level of the whole transcriptome.  - Modified discussion to reinforce our definition of dedifferentiation: At the same time, our analyses question the suggestion that adult, epithelial-derived kidney cancers revert to a fetal state at the whole transcriptome level (i.e., “dedifferentiate”).
5.2	The authors shows interesting examples on how this approach can be used to generate diagnostic clues. While the results are certainly encouraging, only two examples are not enough to tell how accurate and generalizable the method may be for such use (only 2 patients). We would recommend changing the title of the section “Single cell signal provides diagnostic clues” to reflect this, and explain this further in the discussion.	We thank the reviewer for bringing this to our attention. Upon rereading this section, it is clear that we have not explained our approach in this section sufficiently well. In this section, we first begin by taking all our data and asking how accurately can we identify the tumor of origin using only the results of cellular signal analysis. The results of this analysis are shown in (Fig. 7A,B) and demonstrate that this approach can identify the tumor type of a sample with good sensitivity and specificity. In this revision, we provide further validation by showing the correct tumor type can be recovered applying the same approach to non-primary Wilms tumors not included in our main analysis. We have rewritten this section to make clear that it is this analysis that forms the basis for our claim that cellular signal analysis can provide diagnostic clues. An overarching finding of our study was that each tumor type possesses a particular pattern of cellular signals that were uniform in, and specific to, bulk transcriptomes from individual tumor types. Accordingly, cellular signal assessment of bulk transcriptomes may provide sensitive and specific diagnostic clues. To test this proposition, we assessed how accurately the tumor type of each sample in our data could be

		determined based only on its cellular signals. We found that the prevalence of the most common cellular signal for each type could be used to infer each bulk transcriptomes tumor type (Fig. 7A-B, S15). As further validation of this approach we applied this approach to non-primary Wilms tumors (metastatic, secondary) that were excluded from our main analysis. All were correctly identified as childhood tumors and had cellular signals consistent with Wilms tumor (Fig. S16). The two case studies examined in detail are the two samples in our cohort that have a high degree of ambiguity using standard clinical approach and where the additional information that cellular signal analysis can provide is of most interest. That is, the two case studies simply provide a concrete demonstration of when this additional information can be clinically useful
5.3	In the discussion, the authors state that they observe “Remarkably uniform cellular signals”. It is not clear to which results exactly the authors refer to when making this statement. It would also help if the authors could give and discuss specific examples of such features from the data.	Several of the diseases we study here show a large diversity in their clinical course (e.g. Wilms tumour). Despite this, the dominant transcriptional signal they exhibit is basically the same across the entire cohort. This points to the “transcriptional base” of a tumour type being essentially the same within a tumour type, suggesting that the clinically important differences within tumour type are driven by relatively small perturbations on this “remarkably uniform” background. We have changed the text in the discussion to make this clearer: A further finding of our study was that within each category, the majority of tumors exhibited remarkably uniform cellular signals. That is, despite a high diversity in clinical outcome, tumors of the same type almost universally had the same dominant cellular signal (Fig. 7A-B). This indicates that there

		are overarching transcriptional features, beyond individual gene markers, that unite tumor entities despite underlying intra- and inter- tumor genetic heterogeneity.
5.4	Are there plans to make this new method to deconvolute signal contribution as a package? In its current state, it is not clear how other scientists may use this method with their own data. Furthermore, what are the requirements and limits of the method? For example, how many cells / samples are required to obtain accurate deconvolution? I think making this tool available will certainly be important for the field!	We would be delighted to make our method available. We had included the code as part of our submission, but as an extra convenience we have created a github page with the code, a basic example, and some documentation on how it should be used. This can be found here https://github.com/constantAmateur/cellSignalAnalysis
5.5	It would be useful to add titles to the plots of figure 7 E and G to indicate what they represent.	Fixed. Thank you.

Reviewer #4 (Remarks to the Author):

Looks good to me. Authors have addressed my concerns.

Reviewer #5 (Remarks to the Author):

I appreciate the changes the authors have made and do not have further comments.